



# The Utah urban carbon dioxide (UUCON) and Uintah Basin greenhouse gas networks: Instrumentation, data and measurement uncertainty

Ryan Bares[1,2], Logan Mitchell[1], Ben Fasoli[1], David R. Bowling[1,3], Douglas Catharine[1], Maria
Garcia[3], Byron Eng[1], Jim Ehleringer[3], John C. Lin[1]

[1]Department Atmospheric Sciences, University of Utah, Salt Lake City, UT, USA
[2]Global Change and Sustainability Center, University of Utah, Salt Lake City, UT, USA
[3]School of Biological Sciences, University of Utah, Salt Lake City, UT, USA

*Correspondence to*: Ryan Bares (ryan.bares@utah.edu)

**Abstract.** The Utah Urban $CO_2$ Network (UUCON) is a network of near-surface atmospheric carbon dioxide ($CO_2$) measurement sites aimed at quantifying long-term changes in urban and rural locations throughout northern Utah since 2001. We document improvements to UUCON made in 2015 that increase measurement precision, standardize sampling protocols, and expand the number of measurement locations to represent a larger region in northern Utah. In a parallel effort, near-surface $CO_2$ and methane ($CH_4$) measurement sites were assembled as part of the Uintah

Basin Greenhouse Gas (GHG) network in a region of oil and natural gas extraction located in northeastern Utah. Additional efforts have resulted in automated quality control, calibration, and visualization of data through utilities hosted online (https://air.utah.edu). These improvements facilitate atmospheric modeling efforts and quantify atmospheric composition in urban and rural locations throughout northern Utah. Here we present an overview of the instrumentation design and methods within UUCON and the Uintah Basin GHG networks as well as describe and

report measurement uncertainties using a broadly applicable and novel method. Historic and modern data described in this paper are archived with the National Oceanic and Atmospheric Administration's (NOAA) National Centers for Environmental Information (NCEI) and can be found at https://doi.org/ 10.7289/V50R9MN2 and https://doi.org/10.25921/8vaj-bk51 respectively.

## 1 Introduction

Increasing atmospheric carbon dioxide ($CO_2$) caused by anthropogenic fossil fuel combustion is the primary driver of rising global temperatures (IEA, 2015), which has led to international commitment to reduce total carbon emissions. This includes the recent Paris Climate Agreement (Rohdes, 2016) which provided a framework for countries and sub-national entities to make carbon reduction commitments. Cities are playing an increasingly prominent role in these efforts including Salt Lake City, which has committed to a 50% reduction in carbon

emissions by 2030 and an 80% reduction by 2040, relative to the baseline year of 2009 (Salt Lake City Corporation, 2016). Progress on emissions reduction efforts can be evaluated with accurate greenhouse gas measurements to provide trend detection and decision support for urban stakeholders and policymakers who are assessing progress on their mitigation efforts.



Data used to study modern near-surface atmospheric $CO_2$ mole fraction come from a variety of sources.
Flask-based sampling networks such as the one led by NOAA-Earth System Research Laboratory (Tans & Conway 2005; Turnbull et al., 2012) offer long-term, globally representative records of several atmospheric tracers but can be expensive to operate, create temporally sparse datasets, and often do not capture intra-city signals. To supplement flask collection efforts, multiple tall tower greenhouse gas networks exist in North America (Zhao et al., 1997; Bakwin et al., 1998; Worthy et al., 2003; Andrews et al., 2014). These networks make continuous, calibrated $CO_2$
measurements and help to fill in the temporal gaps inherent to flask-based collection. However, by design tall towers are often located away from highly populated regions. Distance from urban emissions make tall tower measurements an invaluable tool for regional scale analysis and background estimates, but similar to flask collection networks they are unable to capture intra-city emissions signals.

While the majority of anthropogenic $CO_2$ emissions occur as a result of human activities in urban areas
(Hutyra, 2014; EIA, 2015), most $CO_2$ monitoring sites are located away from urban sources to measure well-mixed concentrations. Thus, long-term $CO_2$ concentrations measured within urban areas are rare. Established in the year 2001 (Pataki et al., 2003), the Utah Urban $CO_2$ Network (UUCON) is the longest running multi-site urban-centric $CO_2$ network in the world (Mitchell et al., 2018) (Fig. 2).

UUCON collects near-surface data used to (a) understand spatial and temporal variability of emissions
(Pataki 2003; Pataki et al., 2005; Mitchell et al., 2018; Bares et al., 2018), (b) evaluate the accumulation of pollutants during complex meteorological conditions (Pataki et al, 2005; Gorski et al., 2015; Baasanbdorj et al., 2017; Bares et al., 2018, Fiorella et al., 2018), (c) develop and improve atmospheric transport models (Strong et al., 2011; Nehrkorn et al., 2013; Mallia et al., 2015), (d) validate emissions inventory estimates (McKain et al., 2012; Bares et al., 2018), (e) investigate relationships between urban emissions and air pollution, (Baasandorj et al., 2017;
Mouteva et al., 2017; Bares et al., 2018), (f) and inform stakeholders and policymakers (Lin et al., 2018).

To leverage available infrastructure in urban environments and to increase the signals of intra-urban emissions, measurement sites within UUCON are located closer to ground level (Table 1) than tall tower measurement sites. Building-to-neighborhood-scale anthropogenic and biological fluxes contribute more strongly to the UUCON measurements relative to remote-location flask and tall tower observations. Studies comparing tower to
near surface measurements in urban environments have identified an "urban canopy" effect that leads to elevated nocturnal concentrations relative to higher above ground level (agl) measurements (Moriwaki et al., 2006). Thus, the near-surface UUCON data are applicable to research efforts, such as near field emission studies and smaller spatial scale analysis (~1 km$^2$ footprint, Kort et al., 2013) as well as mapping of spatial and temporal heterogeneities in urban emissions and intra-city modeling efforts (Fasoli et al., 2018).
In recent years, cities around the world have launched efforts to establish urban near surface $CO_2$ monitoring observatories for top-down emission estimates and for modeling validation efforts similar to the UUCON network (Mitchell et al., 2018). These cities include Los Angeles (Duren and Miller, 2012; Newman et al., 2013; Verhulst et al., 2017), Indianapolis (Turnbull et al., 2015), Paris (Breon et al., 2015; Staufer et al., 2016), Rome (Gratani and Varone, 2005), Davos, Switzerland (Lauvaux et al., 2013), Portland (Rice and Nostrom, 2011),
and Boston (Sargent et al., 2018), among others (Duren & Miller, 2012). In these studies the number of



measurement locations utilized is fewer than 5, many using a single measurement location to quantify city-wide $CO_2$ variability, with the notable exceptions of Indianapolis (Turnbull et al., 2015) and Los Angeles (Verhulst et al. 2017). While each of these studies employs somewhat similar measurement techniques, UUCON is unique in its length of record (Mitchell et al., 2018).

Starting in 2015, the University of Utah deployed a network of high frequency, high precision analyzers aimed at continuously measuring $CO_2$ and $CH_4$ from areas in eastern Utah where oil and natural gas extraction activities are prevalent (Figs. 2 and 3). These efforts were built on work previously conducted estimating fugitive $CH_4$ emissions (Karion et al., 2013) and the resulting local air quality problems (Edwards, 2013; Edwards et al., 2014; Koss et al., 2015). These methods have also been adopted at two UUCON sites to add $CH_4$ observations to the

urban $CO_2$ record.

     The aim of this paper is to describe the UUCON and Uintah Basin GHG measurement procedures, site locations and data structure with sufficient detail to provide documentation for analyses using these datasets, thereby serving as an in-depth methods reference. Furthermore, we developed a novel method for exploring and quantifying the measurement uncertainty which was used to analyse the performance of the network over multiple years, to

provide insight into appropriate applications of the data, and to explore differences in data collection methods and instrumentation types. This unique method does not require the presence of a target tank with in the dataset, allowing for it to be broadly applicable to many trace gas and air quality datasets that are limited to calibration information alone.

## 2 Network Overview

Currently, UUCON is comprised of nine sites that are dispersed across northern Utah (Fig. 1, Table 1). Six of the sites are in the Salt Lake Valley (SLV), the most heavily populated area of Utah with over 1 million residents as of this writing and where Salt Lake City, the state capital is located. The SLV is surrounded by mountains on all sides except for the northwestern part, where it borders the Great Salt Lake (Fig. 1). Sites in the SLV span multiple characteristics and land uses including residential, mid-altitude, mixed-use industrial, and rural. Two additional sites

are located in the rapidly developing surrounding Heber and Cache Valleys, where the towns of Heber City and Logan are located. Both sites in the developing surrounding valleys are located in predominately residential or mixed commercial zones. In addition to the valley-based sites, a nearby high altitude $CO_2$ monitoring station (HDP), originally started and maintained by the National Center for Atmospheric Research as part of the Regional Atmospheric Continuous $CO_2$ Network in the Rocky Mountains (RACCOON; Stephens et al., 2011), has monitored

$CO_2$ levels that serve as a regional background. The HDP site transitioned into the UUCON network in Fall 2016, at which time $CH_4$ observations were added, and continues to be maintained by the University of Utah.

     Additionally, the University of Utah maintains a network of three greenhouse gas (GHG) monitoring sites in the Uintah Basin of eastern Utah, where energy extraction is taking place, measuring both $CO_2$ and $CH_4$ (Figs. 1, 2, & 3; Table 1). The measurement techniques used in the Uintah Basin GHG network differ from UUCON in

several ways including the use of a different analyzer and will be discussed in detail in Sections 2.3 and 4.1. These



methods have been adapted at two sites within the UUCON network (HDP and UOU) in an effort to add more GHG measurements ($CH_4$) to the data record.

### 2.1 UUCON Instrumentation

Starting in 2001, researchers at the University of Utah deployed Li-6262 (Li-Cor inc., Lincoln, NE) infrared gas analyzers (IRGA) to measure $CO_2$ mole fractions in the SLV. Previous papers have described various different phases of the initial measurement sites (Pataki et al., 2003, 2005, 2006, 2007) (Fig. 2). This paper will focus on the methods and instrumentation developed in 2014 and implemented across the network by summer of 2016, as well as the methods developed for the Uintah Basin GHG network (Fig. 3). Much of the equipment and materials used during the original phase of the network informed the selection of materials for the 2015 overhaul;

however, all components with the exception of the IGRA's were replaced or rebuilt completely and the methods driving these components are significantly different or improved compared to the original design. Additional components were added to increase the functionality, stability and the maintenance of measurement sites (Fig. 4).

A continuous flowing, high frequency method was developed for all locations. At each site, sample gas is continuously passed through the sample cell of a Li-6262 to measure $CO_2$ and $H_2O$ mole fractions (Fig. 4, section

2.1.1). A small positive pressure is maintained throughout the analyzer and measurement system to make the identification of leaks easier and to reduce the impact on the accuracy of data in the event of a leak. Data is recorded and stored as 10-second integrations of 1-second scans.

The decision to change from the historical method, a non-continuous 5 minute collection, to the current continuous 10-second data collection was an effort to better capture higher frequency variations in observed values

that could indicate near-field emissions. Additionally, high frequency data allow for easier identification of "contamination" of the measurement site from highly localized emissions (e.g., furnace, car) that can affect the signal at a site. Finally, while current atmospheric models are limited in their ability to address near field emissions effectively, advances in modeling efforts and computational resources makes this type of analysis feasible in the near future (Fasoli et al., 2018). Thus the high frequency collection of UUCON data is in anticipation of future

model and analysis needs.

Multiple additional measurements are made to ensure the site's reliable performance, increase measurement accuracy, and to assist in identifying instrumentation problems when they arise (section 2.1.7). All data are downloaded and displayed in real time on a public website (http://air.utah.edu) to reduce the time required to identify equipment failure and to provide public outreach. Pressure and water vapor broadening corrections, as well

as data calibration, are performed post data collection and will be described in depth later (section 3). Two sites in the UUCON network, UOU and HDP (Table 1), host an Ultra Portable Greenhouse Gas Analyzer (915-0011, Los Gatos Research, San Jose, CA) onsite. These sites use similar methods as those instrumented with the Li-6262 and will be discussed in-depth in section 2.2.

Lastly, the historic measurement design of UUCON included a 5-liter mixing buffer, which provided a

physical mechanism for smoothing atmospheric observations and reducing instances of large deviations in



observations. After moving to a continuous flow design, the buffer has been removed to enable us to measure high frequency variations. Smoothing can still be achieved at the post-processing and data analysis stages.

### 2.1.1 Infrared Gas Analyzer (IRGA)

A Li-6262 infrared gas analyzer (IRGA) continuously measures $CO_2$ and $H_2O$ mole fraction. The IRGA contains

two optical measurement cells and quantifies $CO_2$ concentration as the difference in absorption between the two cells with a 150um bandpass optical filter centered around 4.62um. To achieve a concentration measurement relative to zero, a $CO_2$ free gas (ultra-high purity nitrogen) is flowed through the reference cell while the gas of interest in passed through the sample cell (Fig. 4).

### 2.1.2 Datalogger

A Campbell Scientific datalogger (CR1000, Campbell Scientific, Logan, UT) acts as both a measurement interface and control apparatus at each site. The datalogger records serial data streams from the gas analyzer, as well as analog voltage measurements from the gas analyzer and all additional periphery measurements. Periphery measurements include: flow rates, room temperature, sample gas pressure, sample gas temperature, and sample gas relative humidity. Several sites have additional air quality measurements that are recorded by the CR1000 (Table 1)

which are not discussed here. The CR1000 is also responsible for driving the calibration periphery that introduces standard gases to the IRGA every two hours (Sec. 2.1.7).

### 2.1.3 Pump and Sample Loop Bypass

Atmospheric sample air is pulled from the inlet to the analyzer using a 12-volt swinging piston pump (UMP850KNDC-B, KNF Neuberger Inc., Trenton, NJ) that provides a reliable flow of 3 L/min. This flow rate is

substantially higher than the 0.400 L/min sample flow rate selected for use at the analyzer. Thus, the pump is located upstream of the manifold where a sample loop bypass provides an alternative exit for unused sample gas. This loop is comprised of at least 9 meters of Bev-A-Line to provide sufficient resistance to the gas so when the manifold is open, gas passes through the mass flow controller and into the analyzer at the desired rate without losing all of the gas to the sample loop bypass (Fig. 4).

### 2.1.4 Relays, Manifold and Valves

Switching from sample gas to calibration gases is achieved using a six position 12-volt relay (A6REL-12, Campbell Scientific, Logan, UT), triggered by the datalogger at a known interval, connected to a six-port gas manifold (Ev/Et 6-valve, Clippard Instrument Laboratory, Inc., Cincinnati, OH) housing 12-volt Clippard relay valves (ET-2-12, Clippard Instrument Laboratory, Inc., Cincinnati, OH). Thus, when the program on the datalogger specifies, the

CR1000 triggers a relay closing the sample valve and introducing a gas of known $CO_2$ mole fraction. Since the maximum number of gases used at each sampling location is five, the unoccupied position on the relay is often used to power the atmospheric sample pump.



### 2.1.5 Mass Flow Controller

A Smart-Trek 50 mass-flow controller (Sierra Instruments, Monterey, CA) is located between the manifold and analyzer to hold the sample flow consistent at 0.400 L/minute (Fig. 4). Flow rates are recorded by analog measurement to the CR1000 to ensure a positive pressure remains consistent, and to help identify measurement issues remotely.

### 2.1.6 Calibration Materials

Each site houses three whole-air, high-pressure cylinders with known $CO_2$ concentrations which are directly linked
to World Meteorological Organization X2007 $CO_2$ mole fraction scale (Zhao and Tans, 2006). Every two hours, the three calibration tanks are introduced to the analyzer in sequence. Each transition of gas begins with a 90 second flush period (ID -99) proceed by a 50 second measurement period, or two hours (minus calibration time) in the case of atmospheric sampling.

The molar fractions of calibration gases are chosen in an effort to span expected atmospheric observations.
Values of the three reference materials are chosen to align with the 5[th], 50[th], and 95[th] percentile of the previous year's seasonal network wide observations (Fig. 5). Utilization of previous observations as a reference allows for a guided estimate of expected observations, thereby allowing for a minimization of interpolation without increasing extrapolation significantly, thus limiting extrapolation bias during calibrations.

In addition to the standard calibration gases, a long-term target tank is introduced to the analyser every 25
hours. This tank is used to quantify performance of the site as well as determining the accuracy of post-processed calibrated data. The interval of 25 hours was selected to ensure that the calibration occurs at a different time each day in order to remove any consistent diel basis, and to prevent the loss of atmospheric observations at a reoccurring time.

Calibration gases are produced in-house using a custom compressor design. 150-liter CGA-590 aluminum
tanks are filled with city air using a high-pressure oil free industrial compressor (SA-3 and SA-6, RIX Industries, Benicia, CA). This system is similar to NOAA-ESRL Global Monitoring Divisions (GMD) system (http://www.esrl.noaa.gov/gmd/ccl/airstandard.html). Water is removed prior to the tanks using a magnesium perchlorate trap to guarantee a dry gas. Tanks are spiked using a ~5,000 ppm dry $CO_2$ tank allowing for a wide range of targeted concentrations depending on the season and expected range of observed atmospheric observations.

Our facility maintains a set of nine standard tanks originally calibrated by NOAA-ESRL's GMD that range from 328 to 800 $\mu$mol mol[-1] (during 2000-2004, directly linked to WMO Primary cylinders). Five of the original laboratory primary tanks were re-measured by GMD in 2011-2012 and were found to be lower than the originally measured $CO_2$ mole fraction by 0.10 to 1.52 $\mu$mol mol[-1].

Laboratory primary tanks (which span 350 – 600 $\mu$mol mol[-1]) are propagated from the above into
"laboratory secondary" tanks using a dedicated Li-7000 (LI-COR Biosciences, Lincoln, NE), and these are used in groups of 5 to calibrate working "tertiary" tanks used in the field. Secondary tanks are replaced as needed; since measurements began, nine secondary tanks have been used. Secondary calibration tanks are periodically re-measured relative to the WMO-calibrated tanks and are generally within 0.5 $\mu$mol mol[-1] of the original



measurement. To assign a known concentration number to tertiary working calibration tanks, each tank is measured
over a minimum of two days, with a minimum of three independent measurements per day.  In a recent laboratory
intercomparison experiment (WMO Round Robin 6), our facilities results were within 0.1 µmol mol[-1] of established
WMO values (https://www.esrl.noaa.gov/gmd/ccgg/wmorr/wmorr_results.php).

The same methods used for developing laboratory primary, secondary and tertiary $CO_2$ tanks were used for
$CH_4$ calibration materials with 5 original tanks spanning from 1.489 – 9.685 µmol mol[-1] $CH_4$. Two of these tanks are
directly tied to the WMO X2004A scale (Dlugokencky et al., 2005). These tanks are propagated into laboratory
standards using a dedicated LGR-Greenhouse Gas Analyzer (Log Gatos Research, 907-0011, San Jose, CA).

A wide array of variables impact the atmospheric CO2 mole fraction of any given urban area including
topography, fossil fuel combustion patterns, biology, and regional background conditions. Thus, intercomparison
and validation of a dataset has primarily been conduced by calibration tank round robin exercises that can tie the
measurements between cities to internationally accepted calibration scales and provides a high degree of confidence
in the validity of the data produced by these measurement networks.

As shown in Figures 2 & 5, winter time $CO_2$ concentrations in the SLV can reach over 650 ppm, with the
95[th] percentile over 550 ppm. As global $CO_2$ concentrations increase in parallel with increasing populations in the
SLV and urban areas of the Wasatch Front (Herbeke et al., 2014), the frequency and amplitude of these highly
elevated observations will increase. Currently the WMO X2007 $CO_2$ scale has a maximum mole fraction of 521.419
ppm. Thus, the current WMO scale may be inadequate for urban observations in the SLV. The urban trace gas
community should consider developing and sharing additional high-quality gas standards with mole fractions more
appropriate to urban observations.

### 2.1.7 Additional Measurements

Three additional measurement sensors were added to the downstream side of the IRGA on the sample line to
provide additional data for identifying equipment failure and to increase the accuracy of dry mole measurements. A
pressure transducer (US331-000005-015PA, Measurement Specialties Inc., Hampton, VA) is located closest to the
analyzer to represent pressures in the sample cell of the IRGA. This data stream is used for post processing pressure-
broadening and water dilution corrections. Uncertainties in the precision and long-term stability of $H_2O$ mole
fraction measurements performed by the IRGA, due to a lack of frequent calibrations of water vapor, led to the
addition of a relative humidity sensor (HM1500LF, Measurement Specialties Inc., Hampton, VA) and a direct
immersion thermocouple (211M-T-U-A-2-B-1.5-N, Measurement Specialties Inc., Hampton, VA) for gas relative
humidity and temperature measurements preformed immediately after the pressure transducer respectively (Fig. 4).
These measurements are utilized to calculate atmospheric $H_2O$ ppm, which is used to calculate $CO_2$ dry mole and
correct for water vapor broadening (section 3.3).

### 2.1.8 Network Time Protocol

Inter-site comparison and modeling applications require a high degree of confidence in the time stamp represented
in data files. To verify the time stamps are consistent between sites and accurate, a network time check is executed





every 24 hours at 00:00 UTC. If the difference between the network clock and the clock on the datalogger is greater than 1000 microseconds, the datalogger clock is reset to match the network clock. All times are recorded in UTC to avoid potential confusion associated with daylight savings.

### 2.2 Uintah Basin GHG Network Instrumentation

The Uintah Basin GHG network utilizes the Los Gatos Research Ultra-Portable Greenhouse Gas Analyzer (907-0011, Los Gatos Research Inc., San Jose, CA), hereafter referred to as "LGR" at all three sites within the network
(Fig. 6). The use of an "off the shelf" analyzer like the LGR compared to system like that generally employed in the UUCON network has both advantages and disadvantages. The barrier of entry of an off the shelf unit is much lower and does not require advanced programming abilities. However, the increase in ease of use results in a decrease in the flexibility of operation, and in some cases the measurement precision decreases (section 4.1).

The Uintah Basin GHG network has supported several recent projects including Foster et al., 2017. In an
effort to minimize differences between the two instrumentation types, measurement frequency, networking, and calibration methods and materials (sections 2.1.6) all follow the same protocols described for the UUCON network with the notable exception of the calibration frequency, which is every three hours as opposed to every two with the Li-6262's.

### 2.2.1 LGR Calibrations

Calibration gases are introduced to the analyser every three hours using three whole-air, high-pressure reference gas cylinders with known $CO_2$ and $CH_4$ concentrations that are directly linked to World Meteorological Organization X2007 $CO_2$ mole fraction scale (Zhao and Tans, 2006) and the NOAA04 $CH_4$ mole fraction scale (Dlugokencky et al., 2005). Calibration gases are introduced using an LGR Multiport Input Unit (MIU-9, Los Gatos Research Inc., San Jose, CA). $H_2O$ mole fractions are calibrated using a Li-Cor LI- 610 dew point generator approximately every
three months.

### 2.2.2 LGR $H_2O$ and Pressure Corrections

Corrections for pressure, water vapor dilution and spectrum broadening for $CH_4$ and $CO_2$ are made on-site by LGR's software and validated empirically by laboratory testing.

### 2.2.3 LGR Additional Considerations

The addition of a target tank, as described in section 2.1.6, would be greatly beneficial for analyzing the long-term performance of each measurement site. However, the current version of the LGR proprietary software that drives the MIU calibration unit lacks flexibility to accommodate a calibration sequence independent of a standard sequence. Thus, the off the shelf nature makes the implementation of this somewhat more difficult.

### 3 Data and Post Processing



Raw data are pulled from each site on a 5-minute interval to the Center for Higher Performance Computing at the University of Utah. Data are then run through an automated calibration and quality assurance program described below and made publicly available at https://air.utah.edu.

### 3.1 Calibrations

Data from UUCON measurement sites with a Li-6262 on site (Table 1) are calibrated every two hours using the
three reference gases outlined in section 2.1.6, while sites with a LGR are calibrated every three hours. Since the Li-6262's are near linear through the range of atmospheric observations and calibration gases, each standard of known concentration is linearly interpolated between two consecutive calibration periods to represent the drift in the measured standards over time (Fig. 7). Ordinary least squares regression is then applied to the interpolated reference values and the linear coefficients are used to correct the observations (Fig. 7). The linear slope, intercept, and fit
statistics are returned for each observation for diagnostic purposes.

### 3.2 Pressure Corrections

Changes in ambient atmospheric pressure can impact the measurement of $CO_2$ mole fraction. Pressure effects can be mathematically accounted for, or minimized or eliminated by maintaining a constant pressure in the optical cavity during calibration and atmospheric sampling periods, as well as calibrating at a high enough frequency that
differences in atmospheric pressure between calibration periods is minimal. We implemented the latter of these two strategies.

### 3.3 Water Vapor Calculations and Corrections

To report dry mole fractions, the presence of water vapor ($H_2O$) must be accounted for. The presence of water vapor impacts measured $CO_2$ mole fraction through both pressure dilution and spectral band broadening. Both of these
effects are corrected for during the post processing of UUCON data. $H_2O$ concentrations are calculated using the relative humidity, pressure and temperature measurements (section 2.1.7) to first determine saturation vapor pressure utilizing the Clausius-Clapeyron relation with Wexler's equation (Wexler, 1976) below:

$$\ln e_s = \sum_{i=0}^{6} g_i T^{i-2} + g_7 \ln(T)$$

(1)

where $e_s$ is the saturation vapor pressure in Pa, $T$ is the temperature in Kelvin and coefficients $g_0 - g_7$ are as follows
respectively: -0.29912729x10$^4$, -0.60170128x10$^4$, 0.1887643854x10$^2$, -0.28354721x10$^{-1}$, 0.17838301x10$^{-4}$, -0.84150417x10$^{-9}$, 0.44412543x10$^{-12}$, 0.2858487x10$^1$.

Vapour pressure ($e$) is calculated using $e_s$ from equation 1:

$$e = e_s \times \frac{RH}{100}$$

(2)



H$_2$O mixing ratio is then calculated by taking the ratio of vapor pressure (*e*) over total atmospheric pressure (*P*) and
converting to parts per million (ppm).

$$H_2O = \frac{e}{P} \times 1000000$$

(3)

Due to the law of partial pressures, the presence of H$_2$O decreases measured CO$_2$ mole fraction. As the amount of
H$_2$O increases, the CO$_2$ mole fraction must decrease for atmospheric pressure to remain unchanged. Using
calculated H$_2$O from equation 1, 2 and 3 we correct for the dilution effect of H$_2$O on the measured atmospheric CO$_2$
using the following equation:

$$CO_{2d} = CO_{2w}\left(\frac{1}{1-H_2O}\right)$$

(4)

where $CO_{2w}$ is the "wet sample" of atmospheric CO$_2$ and $CO_{2d}$ is the dry air equivalent. Given realistic atmospheric
values for the summer in the SLV, 10,000 ppm H$_2$O and 400 ppm CO$_2$, the dilution correction described in equation
4 will result in a positive 4.04 ppm CO$_2$ offset (CO$_{2d}$ = 404.04 ppm).

315       The infrared absorption band utilized by the Li-6262's deployed in the UUCON network is broadened by
presence of H$_2$O resulting in a decrease in the measured CO$_2$ mole fraction. To correct for this effect on the
measured $CO_{2w}$ described in equation 4, we calculated the $CO_{2d}$ in equation 5:

$$Y_C(CO_{2w}) = \frac{a + b \times CO_{2w}^{1.5}}{a + CO_{2w}^{1.5}} + c \times CO_{2w}$$

$$CO_{2d} = CO_{2w}(1 + 0.5H_2O)(1 - 0.5H_2O \times Y_c(CO_{2w}))$$

(5)

where $a = 6606.6$, $b = 1.4306$, and $c = 2.2462 \times 10^{-4}$ and details regarding function $Y_C$ can be found in Li-cor technical
320   documentation (App Note #123).

       Using the same values of 10,000 ppm H$_2$O and 400 ppm CO$_2$, the above equation will result in a -0.66ppm
change. Thus the net correction for both pressure broadening (equation 4) and dilution effect (equation 5) using the
same theoretical H$_2$O and CO$_2$ concentrations results in a 403.3 ppm CO$_2$ dry mole fraction.

**3.4 Data Files**

325   Data are stored at three different levels: raw, QA/QC, and calibrated. Data are stored in monthly files at the native
10-second frequency for all three levels. Raw and QA/QC data files contain an identifier of which gas is currently
being measured with atmospheric air identified as -10, flush periods as -99, and standard concentrations identified as
their known concentration (i.e. 405.06 ppm).

       The lowest level raw data are stored in the same format when pulled from the datalogger at the
330   measurement sites. No periods of data are removed from this level and no corrections or calibrations are applied,
thus remaining totally unaltered.

       The second level of data, QAQC, remains in a similar structure as raw data with a few key exceptions.
First, user specified bad data is removed. A text file containing the periods of "bad data" is maintained for each site,



which is read by automated scripts to remove selected periods. This is a fairly flexible format for removing periods
of suspect data that can be easily updated allowing for quick reprocessing of data. Second, automated quality control
scripts are run and a column of quality assurance flags are added (Table 2). Lastly, calculation of $H_2O$ mole fraction
is performed and $CO_2$ dry mole fraction is calculated as described in section 3.3.

The third and highest level of data, calibrated data, are generated using the QAQC data files. Periods of
invalidated records that fail the automated quality control scripts are removed, and calibrations are applied to all
remaining data.

### 3.5 Sample Sequence

Since all UUCON measurement sites have only one inlet height, atmospheric sampling is continuous between
calibration periods, with no data loss associated with transition periods between sample inlets. During atmospheric
sampling, air is drawn from the inlet and passed through the analyzer continuously where it is identified (ID) as the
numerical value -10 in the raw and QA/QC data files. Every two hours, all three of the calibration materials on site
are introduced to the analyzer in sequence, with a 90 second flush period (ID = -99) to allow for equilibration and
full change-over of the sample cell, followed by 50 seconds of measurement time, resulting in a total of 140 seconds
per calibration gas. Figure 7 shows the transition from atmospheric air to a standard gas and the time required to
reach equilibration. Every 25 hours, a target tank is introduced half way through the hour (i.e., 07:30) using the same
sequence described above, but treated as an unknown and not utilized in the calibration routine described in section
3.1.

### 4 Calculating Measurement Uncertainties

A critical feature of any atmospheric measurement system is an assessment of the system's associated
measurement uncertainty. A comprehensive analysis of greenhouse gas measurement uncertainties has been
described for the NOAA tall tower network (Andrews et al., 2014) and for the LA Megacities project (Verhulst et
al., 2017). Here we have not estimated exhaustively every possible error source. Instead, we have focused on
creating a running uncertainty estimate through time that is similar to the approach taken in the INFLUX project
(Richardson et al., 2017) that does not include an uncertainty estimate for uncertainties from water vapor, calibration
scale reproducibility, or analyzer precision. These uncertainties are small compared to the running uncertainty
estimate and could be estimated in future work.

One method for estimating measurement uncertainties is to use a validation reference gas tank, or "target
tank" ($U_{TGT}$). The target tank is similar to the other calibration gas tanks, but it is not used to calibrate the data and is
also sampled at a lower temporal frequency (once every 25 hours; Sect. 2.1.7). An example of the target tank
measurement is shown in the right panel of Figure 7, where the target tank was measured at 07:30 UTC. The target
tank measurements are treated as an unknown and calibrated (section 3.1). The absolute value of the difference
between the post-calibrated and known values of the target tank is then calculated. We smoothed the absolute
difference time series by convolving it with an 11-point Gaussian window derived according to:

$$e^{-\frac{1}{2}\left(\alpha\frac{n}{(N-1)/2}\right)^2}$$



(6)

where α is 2.5, $N$ is the number of points (11), and $n$ is the sequence between $(N-1)/2 \leq n \leq (N-1)/2$. Prior
studies have also used smoothed target tank values to represent measurement uncertainty through time; however,
each research group has used a different method. For instance, in the NOAA tall tower network, the 1σ absolute
value of the difference between the measured and known target tank mole fractions was calculated across a 3-day
processing window (Andrews et al., 2014). In the LA Megacities project, the root mean square error (RMSE) across
11 target tank measurements (measured every 25 hours) was used (Verhulst et al., 2017). Finally, in the INFLUX
project a running standard deviation of the absolute value of the difference between the measured and known target
tank mole fractions over 30-days was used (Richardson et al., 2017). While these approaches differ in their details,
each represents an assessment of $U_{TGT}$ through time. Future work could examine how the different target tank-based
uncertainty estimates compare to each other and how they affect atmospheric inversion estimates.

Within the UUCON network, target tanks were incorporated into the experimental design in July 2017 at
all of the sites with a Li-6262 analyzer, while sites equipped with a LGR analyzer did not host a target tank, as of
this writing. Thus, to estimate the measurement uncertainty at the LGR sites as well as at Li-6262 sites prior to the
deployment of the target tanks, an alternative measurement uncertainty method was needed. We produced a method
that takes the calibration gas measurements at time $t$, treats them as pseudo target tanks, and interpolates the
calibration gas measurements between the prior ($t-1$) and next ($t+1$) calibration periods to derive a slope and
intercept at time $t$ that is then used to calculate the calibrated mole fraction mixing ratios of the pseudo target tanks
and derive an uncertainty estimate. An example of this process is shown in Figure 8 for the calibration on Nov 27,
2017 at 18:00 UTC at the IMC site. The calibration gas measurements were interpolated between 16:00 ($t$-1) and
20:00 ($t$+1) and used to obtain an interpolated slope and intercept at 18:00 ($t$) (blue dashed line and triangles in Fig.
8a). The interpolated slope and intercept can be compared to the actual values obtained from the usual calibration
procedure (orange circles). The blue dashed line illustrating the interpolation procedure is only shown between
16:00 and 20:00 for clarity, but this process was repeated for each calibration time period. The interpolated slope
and intercept were then used to calibrate the pseudo target tank measurements at $t$ (blue triangles in Fig. 8b). The
RMSE between the calibrated and known values of the three pseudo target tanks was then calculated (grey circles in
Fig. 8d). Since the RMSE can vary substantially between calibration points, we smoothed it by convolving it with an
11-point Gaussian window to yield the pseudo target tank uncertainty, or $U_{pTGT}$ (blue circles in Fig. 8d). For this
example at 18:00, the interpolated calibration intercept resulted in a relatively large deviation of the calibrated
pseudo target tank mole fractions from their known values that then resulted in an elevated RMSE. The elevated
RMSE from this calibration point then persists for several calibration periods (hours) in the smoothed $U_{pTGT}$.

Once $U_{pTGT}$ was calculated, we compared it to the traditional $U_{TGT}$ over time at the IMC site (Fig. 9). For
reference, the yellow shaded region in Figure 9 is the time period shown in Figure 8. In July-August 2017 at IMC
there was a bias in the calibrated target tank mole fractions that similarly affected the pseudo target tank RMSE
values (Fig. 8d). In September 2017 the third calibration tank was removed from the site for a month and the RMSE
values of both metrics improved. Finally, in October 2017 a third calibration tank was re-installed and there was
again a bias in the target tank and pseudo target tanks. The close fidelity through time between the $U_{pTGT}$ and $U_{TGT}$



metrics provides confidence that $U_{pTGT}$ serves as a robust estimate of measurement uncertainty that is similar to what can be obtained with a traditional target tank. Finally, Figure 10 shows the entire $CO_2$ $U_{pTGT}$ and $U_{TGT}$ record at all of the sires, while Figure 11 shows the entire $CH_4$ $U_{pTGT}$ record. The $U_{pTGT}$ is reported in the hourly averaged data files as our estimate of measurement uncertainty.

The average absolute difference between $U_{pTGT}$ and $U_{TGT}$ at all sites in the UUCON network was 0.18 ppm
$CO_2$, suggesting this metric is representative of a more directly measured uncertainty metric like $U_{TGT}$.

### 4.1 Instrument Differences and Uncertainties

A unique aspect of the UUCON and Uintah Basin networks is the use of two different instruments to measure $CO_2$. This allows the ability to directly compare instrument performance during extended field operations. Table 3 shows the uncertainty metrics described in section 4 and in Figures 8, 9, 10 and 11. Additionally, the
precision of the instruments ($U_p$) at each site is reported as an average value of the standard deviation ($1\sigma$) of the post calibrated values for each individual calibration gas introduced to the analyzer since the overhaul of the site, as well as the data recovery rates for each site. Site to site variability in $U_{pTGT}$ ranges from 0.17 to 0.70 ppm $CO_2$ within the UUCON network, with the highest observed uncertainty at sites with more limited environmental controls. Sites equipped with a LGR ranged from 0.17 to 0.32 $CO_2$ ppm (1.8 to 3.3 ppb $CH_4$), with a mean across all sites of 0.27
ppm $CO_2$ (2.8 ppb $CH_4$).

Our reported average $CH_4$ $U_{pTGT}$ uncertainty value of 2.8 ppb is notably higher than those reported by other groups quantifying measurement uncertainty, including Verhulst et al., 2017 which reported a value of 0.2126 ppb uncertainty as estimated using the post-calibrated target tank residuals integrated over 10 days of observations, and a total $CH_4$ uncertainty ($U_{air}$) of 0.7224 ppm from measurements using a Picarro G2301 (Picarro Inc., Santa Clara,
CA). Our higher reported values are likely the result of both the use of a different analyzer than a Picarro, as well as the fact that our uncertainty estimates are based on an interpolation between non-sequential calibration periods and not a directly measured target tank.

It is notable that in all but one instance that the precision ($U_p$) of the Li-6262s $CO_2$ is twice as precise than the LGRs (Table 3), and the one instance is at DBK which experiences larger temperature ranges, despite the Li-
6262s being ~20 years older than the LGRs. Additionally, the uncertainty and data recovery rates between the two instrument types are highly comparable.

The highly similar $CO_2$ metrics observed between the two instrumentation types suggests that the most significant advantage of the more modern direct absorption LGR's is the addition of a second gas species measured, methane ($CH_4$) in this instance, especially at sites with well-regulated climate controls.

**5 Data Availability**

All data described in this paper are archived with the National Oceanic and Atmospheric Administration's (NOAA) National Centers for Environmental Information (NCEI) and can be found at https://doi.org/10.7289/V50R9MN2 and https://doi.org/10.25921/8vaj-bk51.



## 6 Conclusions

440       As the global effort to reduce greenhouse gas emissions transitions from commitment to policy measures, greenhouse gas measurement networks provide a means for evaluating progress. The UUCON network is an example of an urban $CO_2$ network well suited for this application due to its long-term duration, precision, and spatial distribution (Mitchell et al., 2018). With high data recovery rates and low average measurement uncertainty ($U_{pTGT}$) of 0.37 ppm $CO_2$, the network produces data suitable for a range of scientific and, potentially, policy applications.

Additionally, there is increasing interest in performing cross-urban comparisons between different urban environments. Given the reported measurement uncertainties, the frequency of calibrations and the tractability to international working scales, these data are well situated for this application.

      The overhaul of instrumentation and design documented in this paper has resulted in a robust network of reliable data, with additional measurements to remotely identify when problems arise as well as increase the

precision of the data. The standardization of materials and measurement protocols at all locations has significantly lowered the barrier of entry for maintenance of the sites.

      The addition of target tanks at multiple sites in 2017 allows for the calculation of continuous uncertainty metrics. From those metrics, an interpolation method was developed allowing for uncertainty estimates of sites and networks where a target tank is not available. While this method likely results in overestimation in the uncertainty,

this novel method for estimating uncertainty nonetheless provides useful insight into the quality of data produced at individual sites and is broadly applicable to any atmospheric trace gas or air quality dataset that contains calibration information.

      The use of the interpolated uncertainty metric, as well as the calculation of the standard deviation of calibration measurements in the field, identified limited differences between the two measurement techniques used

in the UUCON and Uintah Basin GHG networks.

      Targeted reductions in the emissions of other greenhouse gases, primarily $CH_4$, will require similarly distributed measurement networks for validating reduction progress and tracking emissions, both in urban areas and regions of oil and natural gas extraction.  With three years of continuous operation to date, and relatively low measurement uncertainty (2.8 ppb $CH_4$) the Uintah Basin GHG network serves as a good example of a greenhouse

gas network with simultaneous measurements of $CH_4$ and $CO_2$. With comparable precision and reliability as those reported in UUCON, but with the added benefit of two measurement species, the measurement techniques deployed in the Uintah Basin GHG network have been expanded into a few urban locations within the UUCON network.

### Acknowledgements

This research was supported by the National Oceanic and Atmospheric Administration (NOAA) grant NA140AR4310178 and NA140AR4310138. The authors would like to thank Dr. Britton Stephens and the National Center for Atmospheric Research for establishing the HDP site, Dr. Seth Lyman and Utah State University-Vernal for their continued support of the Uintah Basin GHG network, and The Stable Isotope Ratio Facility for Environmental Research (SIRFER) at the University of Utah for their commitment to UUCON.  We would also like

to thank the following hosting institutions: Draper City and the Salt Lake County Unified Fire Authority, Rio Tinto



Kennecott, Snowbird Ski Resort, The Salt Lake Center of Science Education, Intermountain Health Center, Utah State University Logan, Utah State University Vernal, Wasatch County Health Department, and the Utah Division of Air Quality. All data used in this analysis are available upon request from the corresponding author or can be downloaded at the U-ATAQ's data repository at https://air.utah.edu/data/.



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




**Table 1:** Site Characteristics of the UUCON (top) and Uintah Basin GHG network (bottom). Historic sites that have been relocated are not listed. Details regarding instrument types and measurement details for species other than $CO_2$ and $CH_4$ are not covered in this paper but currently measured at a subset of sites include Carbon Monoxide (CO), Ozone (O3), Fine particulate Matter ($PM_{2.5}$) and Nitrogen Oxides ($NO_x$).

| Site Code | Site Name | Latitude | Longitude | Elevation (m) | Inlet Height (m agl) | Species | Start Year | Overhaul Year | Instrument | Land-Use |
|---|---|---|---|---|---|---|---|---|---|---|
| UOU | University of Utah | 40.7663 | 111.8478 | 1,436 | 36.2 | $CO_2$, $CH_4$, CO, $O_3$, $PM_{2.5}$, $NO_x$ | 2001 | 2014 | LGR UP-GGA | Mixed residential commercial |
| SUG | Sugarhouse | 40.7398 | 111.8580 | 1,328 | 3.86 | $CO_2$, $PM_{2.5}$ | 2005 | 2015 | Li-6262 | Residential |
| IMC | Intermountain Medical Center | 40.6602 | 111.8911 | 1,316 | 66.0 | $CO_2$ | 2016 | NA | Li-6262 | Commercial |
| RPK | Rose Park | 40.7944 | 111.9319 | 1,289 | 3.25 | $CO_2$ | 2009 | 2015 | Li-6262 | Residential |
| DBK | Daybreak | 40.5383 | 112.0697 | 1,582 | 5.05 | $CO_2$, $PM_{2.5}$ | 2004 | 2015 | Li-6262 | Rural sagebrush steppe |
| HDP | Hidden Peak | 40.5601 | 111.6454 | 3,351 | 17.1 | $CO_2$, $CH_4$ | 2006 | 2016 | LGR UP-GGA | High Elevation / Urban Background |
| LGN | Logan | 41.7616 | 111.8226 | 1,392 | 3.23 | $CO_2$ | 2015 | NA | Li-6262 | Mixed Residential Commercial |
| HEB | Heber | 40.5067 | 111.4036 | 1,721 | 4.20 | $CO_2$ | 2015 | NA | Li-6262 | Residential Developing |
| SUN | Suncrest | 40.4808 | 111.8371 | 1,860 | 4.22 | $CO_2$ | 2015 | NA | Li-6262 | Mid-altitude, Residential |
| FRU | Fruitland | 40.2087 | 110.8404 | 2,024 | 4.04 | $CO_2$, $CH_4$ | 2015 | NA | LGR UP-GGA | Basin Background |
| ROO | Roosevelt | 40.2941 | 110.0090 | 1,585 | 4.06 | $CO_2$, $CH_4$ | 2015 | NA | LGR UP-GGA | Basin Residential |
| HPL | Horsepool | 40.1434 | 109.4680 | 1,567 | 4.06 | $CO_2$, $CH_4$ | 2015 | NA | LGR UP-GGA | Oil and Natural Gas |





**Table 2: Quality Assurance and Control Flags**

| Flag | Descriptor |
|------|------------|
| **-1** | Data manually removed |
| **-2** | System flush |
| **-3** | Invalid valve identifier |
| **-4** | Flow rate or cavity pressure out of range |
| **-5** | Drift between adjacent reference tank measurements out of range |
| **-6** | Time elapsed between reference tank measurements out of range |
| **-7** | Reference tank measurements out of range |
| **1** | Measurement data filled from backup data recording source |



**Table 3: $CO_2$ and $CH_4$ Measurement Uncertainties with Gaussian window target tank method ($U_{p\text{TGT}}$), target tank ($U_{\text{TGT}}$), analyser precision at $1\sigma$ ($U_P$) and data recovery rates from UUCON and Uintah Basin GHG measurement.**

| Site Code | $CO_2$ $U_{p\text{TGT}}$ (ppm) | $CO_2$ $U_{\text{TGT}}$ (ppm) | $CO_2$ $1\sigma$ $U_p$ (ppm) | $CH_4$ $U_{p\text{TGT}}$ (ppb) | Data Recovery Rate |
|---|---|---|---|---|---|
| DBK | 0.70 | 1.04 | 0.04 | NA | 0.82 |
| HEB | 0.18 | 0.38 | 0.04 | NA | 0.81 |
| IMC | 0.35 | 0.49 | 0.03 | NA | 0.71 |
| LOG | 0.17 | 0.51 | 0.04 | NA | 0.85 |
| RPK | 0.47 | 0.38 | 0.10 | NA | 0.83 |
| SUG | 0.31 | 0.26 | 0.04 | NA | 0.80 |
| SUN | 0.41 | 0.54 | 0.05 | NA | 0.73 |
| UOU | 0.37 | NA | 0.08 | 3.3 | 0.91 |
| FRU | 0.28 | NA | 0.13 | 2.7 | 0.86 |
| HDP | 0.17 | NA | 0.10 | 2.0 | 0.77 |
| HPL | 0.24 | NA | 0.08 | 4.2 | 0.77 |
| ROO | 0.18 | NA | 0.10 | 1.8 | 0.81 |




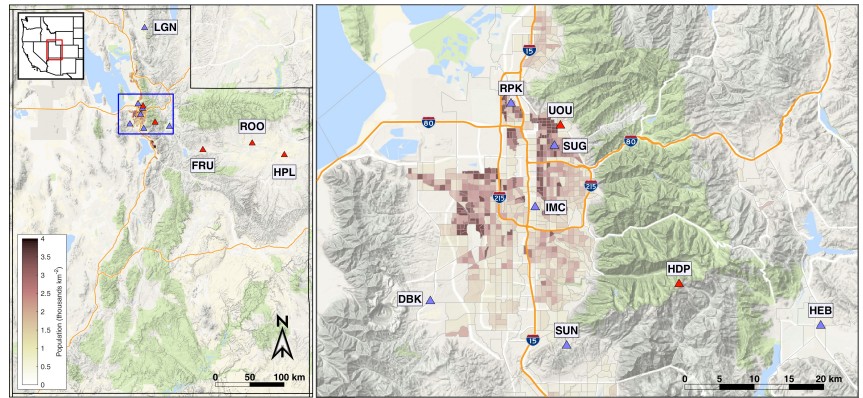

Figure 1: Map showing the location of UUCON and Uinta Basin GHG measurement sites. Left panel shows full distribution of sites in Utah with blue square indicates extent for the right panel. Right panel shows the Wasatch Front and the Salt Lake Valley in detail with population density in thousands per km$^{-2}$. Sites equipped with a Li-6262 identified with blue triangle and sites with an LGR UP-GGA identified with red triangle.



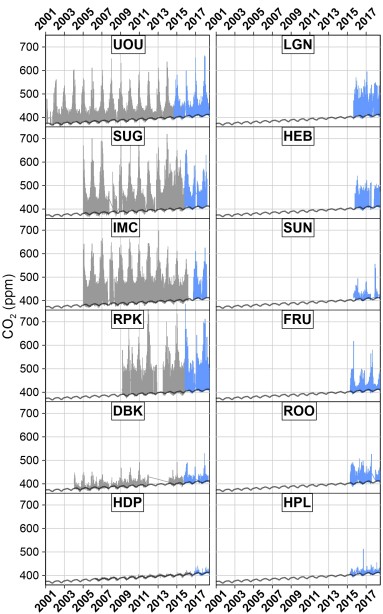

Figure 2: Full record time series of $CO_2$ measurements from the UUCON and Uintah Basin GHG. Measurement techniques and uncertainty covered in this manuscript indicated by blue with historic data represented in grey. Black line represents regional background as described in Mitchell et al., 2018a.




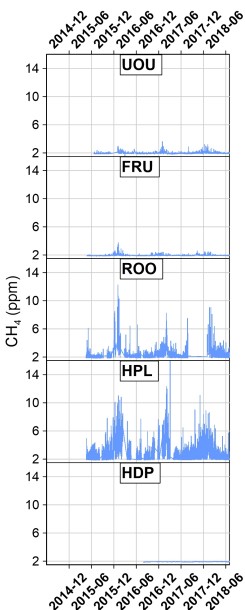

Figure 3: Full record time series of $CH_4$ measurements from the UUCON and Uintah Basin GHG.

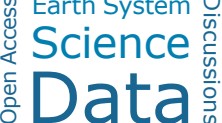

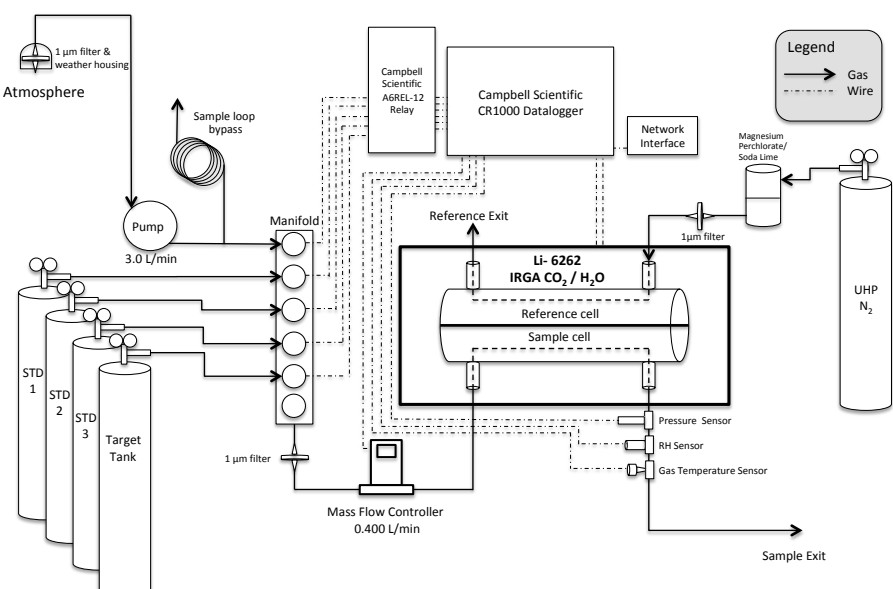

Figure 4: Diagram of UUCON measurement design, not to scale. Sites with this design identified in Fig 1. with blue triangles. STD = Standard Tank.





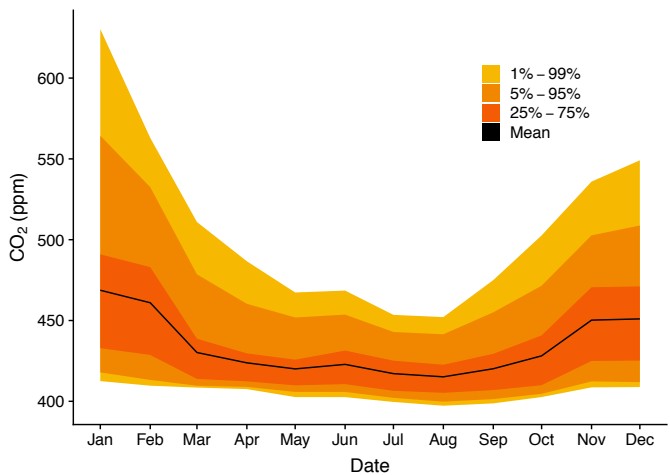

Figure 5: Monthly percentiles of atmospheric observations from SUG over one year, 2017. Note the majority of observations (95th percentile) are greater than 550 ppm $CO_2$, well beyond the current WMO calibration scale.


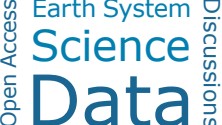



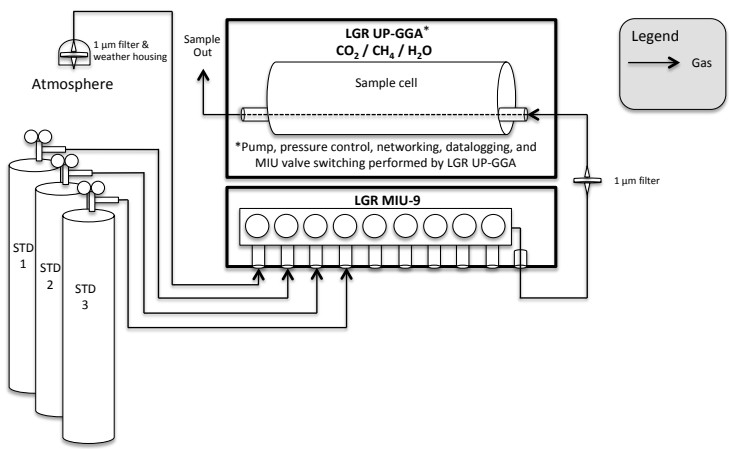

Figure 6: Diagram of Uinta Basin Greenhouse Gas Network measurement design. Sites with this design identified in Fig 1. with red triangles.



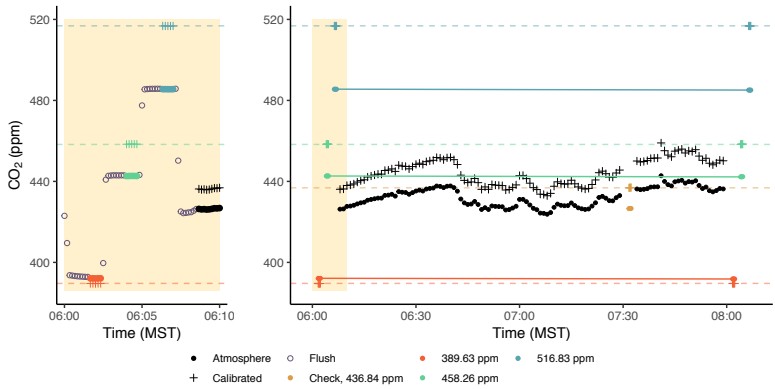

Figure 7: Left panel shows the sequence and timing of a standard calibration period in both the UUCON and Uinta Basin network. Gray open circles indicate the 90 second flushing period observed between each change in gas. Right panel shows a full two hour sample period with calibrations for the UUCON network with linear interpolations, flush periods have been removed. Orange, green, and blue closed circles indicate calibration standard gas and their known $CO_2$ concentration. Yellow closed circle represents the check tank and its known concentration. Black closed circles indicate pre-calibration atmospheric observations which have ben down sampled to one minute averages to reduce over plotting. Plus (+) signs in all colors indicate the post calibrated measurements for the corresponding measurement.



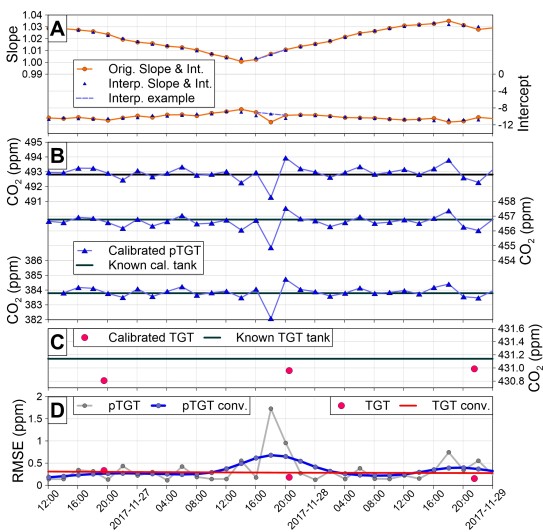

Figure 8: Detailed view of the uncertainty analysis at the IMC site. An example of the interpolation procedure is illustrated for the calibration at 18:00 UTC on November 27, 2017 (see the description in the text). The "pTGT conv." and "TGT conv." curves in panel D are the $U_{pTGT}$ and $U_{TGT}$ uncertainty metrics, respectively.






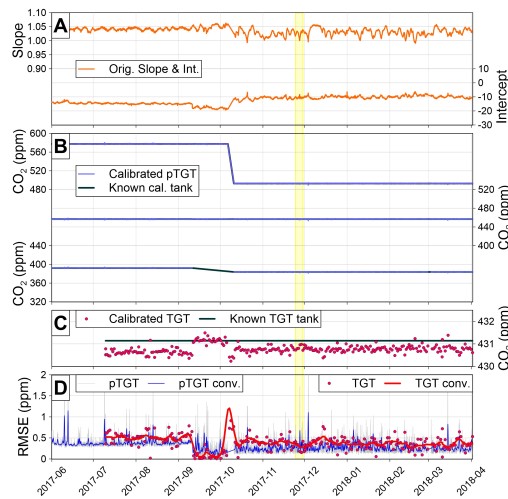

Figure 9: Uncertainty analysis at the IMC site for the time period when a target tank was deployed at the site. The "pTGT conv." and "TGT conv." curves in panel D are the $U_{pTGT}$ and $U_{TGT}$ uncertainty metrics, respectively.



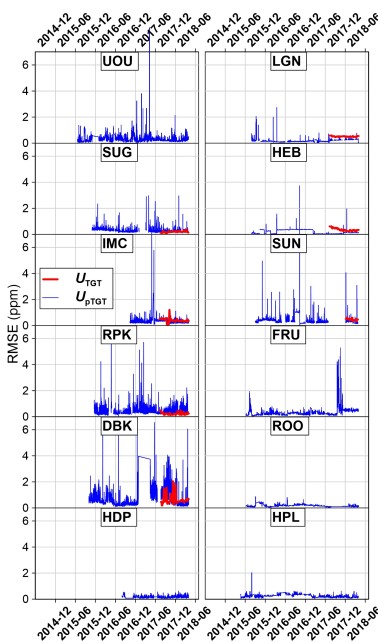

Figure 10: Uncertainty analysis for all of the UUCON sites. The $U_{pTGT}$ and $U_{TGT}$ uncertainty metrics are the same as the "pTGT conv." and "TGT conv" curves in Fig. 8d and 9d, respectively.




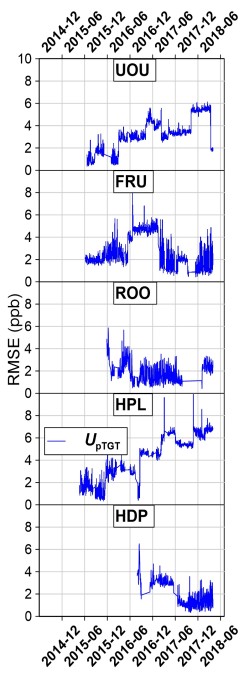

Figure 11: CH$_4$ Uncertainty analysis. All values reported are the $U_{pTGT}$ uncertainty metrics as shown in Fig. 9d.