# Peer review of "The Utah urban carbon dioxide (UUCON) and Uintah Basin greenhouse gas networks: Instrumentation, data and measurement uncertainty"

_Earth System Science Data, 2018_

## Referee Comment (RC1) · Anonymous Referee #1 · 1 Mar 2019

Review of Bares et al., "The Utah urban carbon dioxide (UUCON) and Uintah Basin greenhouse gas networks: Instrumentation, data and measurement uncertainty"

This manuscript is a well-written and organized description of two GHG measurement networks. Overall, most of my comments are extremely minor (wording/spelling/grammar), with only a few more substantial comments.

More substantial comments:

The methods and uncertainties seem sound. Two things could be improved: 1) the pump in the UUCON network is upstream of the analyzer, meaning that the sample (but not the cal gas) goes through the pump first. - the authors should test the output

of the pumps to ensure that Co2 is not compromised or changed by the pump, or add some language addressing this issue. 2) The system does not dry the sample, so the authors should ensure that that the water vapor correction is robust. Although they have done a very nice job explaining how it is applied for the Licor, they state that they do not think it is a large component of the uncertainty but they do not back that up. This should be done for both systems.

Specific comments:

L79. Restate what "These methods" refer to here

L115: IGRA should be IRGA

L118 - reword - what is continuously flowing (data? air?) and what is high-frequency? not the method.

L123 Again, what exactly is meant by "continuous" vs. "not continuous"? Are the 5 minute points not averages and the 10 second data are averages? Or is this just referring to the higher reporting frequency?

L126 I would not put quotes around contamination (not a term being defined here or special usage)

L146. um is not the abbreviation for micrometer, the actual greek mu should be used here

mole fraction should probably be used consistently throughout here rather than switching back and forth with concentration

L159 - I was interested in this pump so I looked it up and cannot find this one - can the authors check the number? The NMP850-KNDC (similar part) seems to be a diaphragm pump, not swinging piston. But perhaps the UMP is a piston? Here it would be desirable (as noted at the top of this review) to indicate whether this pump could affect CO2 concentrations downstream.

L175 L/minute is volume flow - specify standard liters per minute. I would guess at least some if not all the sites are at altitude, so this becomes more important.

L196 Division's (apostrophe)

L200+ I commend the authors on the careful accounting of their standards. It is a little disappointing to hear of drift in the tank of 0.5 ppm however, that seems quite high. (let alone the 1.5 ppm drift in one of the original tanks!).

What is the typical value of the target tank?

L217 subscript 2 in CO2.

L218-221 this paragraph is confusing and a little out of the blue? Is this round-robin between cities occuring or has already been done? (also conduced should be conducted). Or is this a recommendation?

L222, In the previous few paragraphs, the units have been \mu mol mol ˆ{-1}, now switching to ppm - should be done at the top and defined once and be consistent

L226-228. How does this upper limit on the WMO scale affect the calibrations mentioned above which go up to 800 ppm?

L239 H2O mole fraction (in ppm), and missing "fraction" after "dry mole"

L254 including Foster et al.? Perhaps a few more words with the ref in parentheses?

L263 is the NOAA04 scale different from the WMO X2004A scale mentioned earlier for CH4?

L266 section 2.2.2 could be a bit longer - how are these corrections validated in the laboratory? A few more sentences would be useful on this important correction, including an estimate of the error on this (as mentioned below under the uncertainty analysis).

At the top of this section (L250) would be good to mention what gases are measured by the LGR.

Figure 5. Caption is incorrect, as it states that the majority of the data is over 550 ppm, which is certainly not the case if the 95th percentile is 550 then only 5% of the data is above 550!

section 3.2: L 288-290 - clarify whether this is an issue in the Licor and LGR both or only Licor. Does the LGR control cavity pressure? Would the constantly flowing reference gas in the case of the Licor take care of this issue? (perhaps not).

L303: check consistency between British and American spelling of "vapor/vapour"

L305: is this now a mole fraction of H2O or something else (volume fraction, mass fraction? Mixing ratio often refers to mass ratio?)

Section 3.3 Please remind the reader that this is only being done for UUCON because the LGR is doing its own correction interally.

Fig. 7 caption wording: comma should be semicolon? (before flush periods). And here the target is referred to as the check tank, it might be more consistent to refer to it here as the target tank as was done earlier. Also, should say "calibrated", not post calibrated. Or perhaps "post-calibration".

L359: It has not been shown what typical uncertainty due to these might be for this system. Perhaps the authors could indicate what typical analyzer precision is during calibrations for the two instruments (I now see this information is already in Table 1, so you could just refer to this here). More difficult is the water vapor correction - are there any thoughts as to the possible error here, even if just a guess or based on a few tests mentioned earlier where the LGR correction was checked in the lab - what do those tests show? It would be good to be able to back this statement up, that these are likely small compared with the running uncertainty, even if just with some anecdotal information, without a formal estimate of their values.

L361: clarify that this is only available at the UUCON stations

L401. I would think this method would actually not be able to see a bias in the calibration value of a given tank, it only can see whether the calibrations are noisy from day to day, because you are interpolating across calibration periods. For example, if one tank is always biased high due to a bias in the assigned value by the calibration lab, the slope has a bias all the time, and this would not be included in the virtual target method. Unless I am not understanding this right. If I am then it would be good to mention a case where this method does not capture the true full uncertainty that a true target would, or at least provide a sentence as to the limitations. Otherwise, this seems like a good method.

L408 typo: sites

L 409 - Is there an overall difference in the mean? i.e. Is UTGT larger than UpTGT on average? I would think so, but it would be nice to state the mean difference as well as the mean absolute difference here. I now see this in table 1 where it seems that at most sites where both were computed UTGT is greater, but not at all. this could be mentioned here, with a reference to the table.

Table 3: are these the mean uncertainties over some period of time? Please note in the caption.

L416: calibrated rather than post calibrated?

L424: ppm should be ppb for CH4 here.

L435. Is this proper ordering of sections? Data Availability before Conclusions

L454. Why does the UpTGT method likely overestimate uncertainty? There is no other indication of this anywhere in the text.
* * *

---

## Referee Comment (RC2) · Anonymous Referee #2 · 16 May 2019

This manuscript presents the data from the UUCON and Uintah basin GHG networks. The dataset is a valuable addition to the urban GHG research field, and the manuscript overall is well-written in describing the instrumentation and uncertainty analysis associated with this dataset.

I suggest minor revisions to address a few notable questions, namely:

– I feel that one of the most difficult aspects of an urban GHG monitoring network is the delicate balance between being in close to the emissions in the area but not "too close", as in being influenced by emissions in very close proximity to the measurement site that would be difficult to interpret from a modeling perspective. The authors actually

mention this "contamination" in Ln126, as one of the motivations to implement changes to the instrumentation that would better capture high-frequency variations. So, have the authors looked into whether local "contamination" can be identified in the dataset? I feel a flag that identifies periods of potential local "contamination" could be of great value to the potential user of the data, as such screening may require detailed knowledge of the site environement that only the authors could have. If not for this paper, perhaps another study could be done that delves deeper into this issue?

– I find it surprising that the Uptgt results presented in this study seem quite high compared to those from other networks, and also that Figures 10, 11 suggest signficiant variation and trends in this value. The authors do mention differences in instrumentation compared to other networks and influence from the environement where the instruments are operated in as possible explanations for these results, but I do wonder if a more thorough investigation is warranted to really understand this issue.

– Because the calbration/target tanks are dry and the air measurements are made "wet" (without any drying), it seems like uncertainties in the moisture correction would become a significant source of uncertainty in the air measurements, and this uncertainty would not be captured in Uptgt, Utgt. Do the authors have thoughts on this issue? Should some representation of the moisture correction uncertainty be presented to the users of the data?

– I'm a bit puzzled by the relatively poor performance of the LGR instruments compared to the older Licor instruments. Just out of curiosity, I downloaded the FRU station data (the worst instrument in terms of precision), looked at the numbers for a handful of individual cal runs, and the stdev's I see seem much closer to those of the Licor instruments than to the 1-sigma precisions for the LGR instruments reported in Table 3. I would like to ask that the authors check their calculations and make sure they can stand by the results and discussions presented in this study.

Minor comments below:

Ln23: Remove space in the doi link, e.g. ".org/ 10." -> ".org/10."

Ln37: I'm not sure that stating a flask network is "expensive to operate" is a sound scientific statement, I think most would agree that there are substantial costs to any type of measurement effort, and one might even argue that a flask network is a more affordable way to operate a dense, wide measurement network, if you are willing to sacrifice measurement frequency. I would suggest that the statement be removed.

Ln105: Authors should clarify the "These methods" referenced here.

Ln115: IGRA -> IRGA

Ln123: The change from a non-continuous 5-min collection to the 10-sec data collection requires a bit more explanation. My guess is that the 5L mixing buffer mentioned in Ln139 is essentially what the authors refer to here, with the assumption that the residence (mixing) time in the buffer is about 5 minutes? Perhaps the paragraphs here can be rearranged to make this point more coherent?

Ln159: Do the authors mean "NMP" instead of "UMP"? Can the authors clarify the diaphragm and pump head materials for the pump?

Ln162: Bev-A-Line OD/ID's?

Ln167: What is the material and approx. inner volume of the gas manifold? I assume this was addressed by the 90 sec flushing time, but did the authors worry about dead volumes in the manifold? Ditto for the LGR Multiport Input Unit mentioned in Ln263?

Ln179: How long do the calibration tanks last out in the field? What model regulators used?

Ln182: Perhaps "ID -99" can be removed here, as it is a bit confusing without context, and repeated in Ln 328.

Ln194: Can the authors clarify the "150L" capacity of the aluminum tanks? My guess is that the authors refer to the N150 style tanks, with 29.5L internal volume.

Ln198: How was the ∼5000 ppm CO2 spike tank sourced? Is spiking for CH4 done separately using an additional tank?

Ln199: Any special air conditions in which the compressors are operated? Any need for dilution to bring the concentrations down in the tanks, and if so how is this done?

Ln 203: Cause of assigment difference on the tanks? Misassignment or tank drift? Has previous data be updated to reflect the assignment change in the lab standards?

Ln216: Have the authors thought about calibrating CO2 on the LGR as well?

Ln217: Subscript missing in "CO2"

Ln225: The authors should note that NOAA is actually working on extending the CO2 scale to 600 ppm, see presentation from the last GGMT meeting in Switzerland: https://www.wmo.int/pages/prog/arep/gaw/documents/GGMT2017_T04_Hall.pdf

Ln241: Description of the internet stream that provides the network clock is missing. I presume this same internet connection is also used to send the data to the Univ of Utah data server?

LN250: I'm not sure exactly what the authors refer to with "off the shelf". I presume what the authors mean is that the LGR's were largely operated using peripherials from LGR and corrected using LGR's internal algorithms, such that the complete system was "off the shelf". However, I don't think it's fair to characterize the LGR anlyzer itself as more "off the shelf" than the Licor 6262's, I'm sure there are "off the shelf" ways to operate the Licor's as well. I suggest the authors clean up the message here. Also, can you clarify whether the LGR data is calibrated from the internal software, or instead worked up in post processing like the Licor data?

Ln260: Just to make sure, the same protocol of 3 cal gasess that range the actual atmospheric data, prepared in-house?

Ln273: Reword sentence to simply say it wasn't implemented.

Ln274: Can the authors clarify how much of the description in Section 3 apply both to the Li-6262's and the Lgr's, as opposed to only the Li-6262's?

Ln336: Can the authors discuss what checks are implemented in the "automated quality control scripts"?

Ln374: 22 hours, according to Verhulst et al. 2017

Ln395: I don't see blue circles in Figure 8d, but I do see gray circles with a blue line going through them. . .

Ln400: I suggest that the explanation of the yellow shaded region on this line be repeated on the captions for figure 9.

Ln401: "calibrated target tank mole fractions", I initially understood that to mean the concentrations assigned to the target tank from the central lab. Would it be a bit more clear to say "on-site assignment of the target tank"?

Ln402: Can you clarify the "third calibration tank"? Was it the tank with the highest concentration?

Ln402: Not sure why Figure 8d is referenced here?

Ln404: After the "third calibration tank" is re-installed, there is a brief jump in TGT RMSE. Is there an explanation for this? Also, there's a gap in the pTGT calculations at the same time, is there an explanation for this?

Ln407: Are the hourly datafiles with the Up-TGT estimates considered an additional data level, compared to discussions in section 3.4?

Ln429-430: This is an interesting observation. I think one important distinction to make here is that the two analyzers were operated with different peripherals, such that the comparison isn't completely apples-to-apples, per se. I'm also somewhat concerned that the LGR precisions are significantly worse than those suggested in LGR data sheets (100 sec 1-sigma precision for CO2 0.05 ppm, CH4 0.3 ppb). While I understand that company data sheets are not to be trusted, I'm surprised that the LGR's show worse precision than the 20 year old Licor's! Have you tried directly replacing the Licor analyzer with the LGR? I understand that the temperature of the lab can have a significant effect on instrument performance, but I would have suspected that most of those effects would concern instrument drift, and that short-term precisions are relatively less affected, so I find the high Up values for the LGR surprising. Can the authors get into a bit more detail on what might be causing these findings?

Ln454: One thing that's clear is that Uptgt is generally higher than Utgt, at least based on eyeballing Figure 10. However, can you be sure that Utgt doesn't underestimate the data uncertainty, due to the fact that the 25hr sampling sequence doesn't fully capture the calibration uncertainties that happen at 3~6hr intervals?

Please check subscripts for $CO_2$, $CH_4$ in the reference list.

Multiple Mitchell et al. 2018 references on list, should clarify in accordance with ESSD style guidelines.

Table 1: Lat/Lon's should specify N, W.

Figure 4: Can the authors clarify the tubing used in the gas connections in the main text?

Figure 7: "Check gas"? Why not "target" tank, just to be consistent?

Figure 9. I'm not sure what 9B is showing here. Also, is there some gray in the background of 9D, and what does that reflect?

Figures 10-11. I'm surprised that there is so much variation in Uptgt, especially for $CH_4$ where it seems like there are clear jumps (likely at cal tank changes) and long-term drift. Have the authors looked into the cause of these uncertainties? For example I wonder if the spikes in RMSE errors correlate with large spikes in the air concentrations, in which case one may suspect a leak or memory effect in the system?

[Figure]

---

## Author Comment (AC1) · 27 Jun 2019

I, along with my co-authors, would like to thank the editorial board and the referee's for the consideration of our manuscript, entitled "The Utah urban carbon dioxide (UUCON) and Uintah Basin greenhouse gas networks: Instrumentation, data and measurement uncertainty." We are grateful for your time and thoughtful comments. We have carefully addressed each comment provided by the referee's below. We feel that these revisions have greatly improved the quality of the manuscript and appreciate the expertise these referee's contributed to the paper.

Sincerely, Ryan Bares

[Figure]
* * *
_____-

Anonymous Referee #1 Comments:

This manuscript is a well-written and organized description of two GHG measurement networks. Overall, most of my comments are extremely minor (wording/spelling/grammar), with only a few more substantial comments.

More substantial comments:

The methods and uncertainties seem sound. Two things could be improved: 1) the pump in the UUCON network is upstream of the analyzer, meaning that the sample (but not the cal gas) goes through the pump first. - the authors should test the output of the pumps to ensure that Co2 is not compromised or changed by the pump, or add some language addressing this issue.

We thank the referee for pointing out the potential for the pump to influence the atmospheric sample. We also note that referee #2 made a similar comment, which emphasizes the need for us to include additional information on our choice of materials and validation of those materials within the design. The pumps used in the UUCON network were selected to minimize any potential absorption of $CO_2$ or other interference with the sample. The diaphragms are made of a PTFE coated EPDM rubber which has been shown to have negligible gas phase absorption. Additionally, multiple tests were performed in the lab and in the field to verify that the location of the pump upstream of the analyzer would not impact the observations. These tests were performed using pressurized cylinders of known concentrations of $CO_2$ that span well beyond those values observed with in the network, 350 – 1000 ppm. Both dry and wet samples were tested to verify that the presence of $H_2O$ in the sample gas would not impact the results. Wet standard gasses were produced by passing the dry known gas through a Licor 610 dew point generator at a know temperature and pressure. Calibration gasses were passed through a pressure controlled Li-6262 with and with out the presence of

an upstream pump and compared. No measureable changes were identified throughout the full range of CO2 values, both wet and dry. Theses results provide us with a reasonable level of confidence that any absorption or interference from the sample pump being located upstream of the analyzer is negligible and beyond the instruments decidable limits.

Language was added to section 2.1.3 describing the pump material in more depth and briefly outlining the tests and results described above. Line 187 of the track changes version of the mauscript now read:

"Since the pump is located upstream of the analyzer there is potential for CO2 to absorb onto the material with in the pump head and interference with the atmospheric sample. The pumps used in the UUCON network were selected to minimize any potential interference with the sample. The diaphragms are made of a PTFE coated EPDM rubber which has been shown to have minimal gas phase absorption. Multiple laboratory and field tests were performed to verify that the location of the pump upstream of the analyzer would not impact the observations. No measureable impacts were identified provide us with a reasonable level of confidence that any absorption or interference from the pump is negligible."

2) The system does not dry the sample, so the authors should ensure that that the water vapor correction is robust. Although they have done a very nice job explaining how it is applied for the Licor, they state that they do not think it is a large component of the uncertainty but they do not back that up. This should be done for both systems.

This is an excellent point made by the referee, and again one echoed by referee #2, underlying the importance of this comment. In an effort to address this comment, we estimate the measurement uncertainty of the water vapor measurements for both the UUCON and the Uintah Basin GHG networks, describe the laboratory tests used to calculate these values, and report the results in both the text in section 4 as well as in table 3.

Additional language was added starting on line 540 in the manuscript to include these numbers. This paragraph reads:

"Water vapor precision was examined using laboratory tests for the UUCON and the Uintah Basin GHG network designs and are reported in Table 3 (Uh2o). Gas from a dry calibration tank of know $CO_2$ mole fraction was passed through a Li-610 dew point generator at a set dewpoint temperature. $H_2O$ measurements were collected by both systems in parallel over a period of 1.5 hours. We calculated the Allan variance to represent the precision of the $H_2O$ measurements regardless of drift over time or other systematic errors. This precision statistic was used to construct a normal distribution of $H_2O$ centered on the mean measured $H_2O$ mole fraction at each site, which is used to estimate the uncertainties in dry air equivalent estimates for $CO_2$ due to $H_2O$ repeatability error using methods discussed in Section 3.3. The $1\sigma$ uncertainty of the $H_2O$ precision results in a mean 0.019 ppm $CO_2$ error (Uh2o) for the UUCON network, and 0.017 ppm $CO_2$ for the Uintah Basin GHG network design. These uncertainties represent a lower bounds for error in $CO_2$ resulting in $H_2O$ measurements as they do not account for errors in $H_2O$ measurement accuracy. Biases in the accuracy of measurements are addressed in the QAQC procedures of the data."

We feel that the addition of this information, at the suggestion of both referee #1 and #2, was a significant contribution to the manuscript and we are grateful to both for their suggestion to include this information.

Specific comments:

L79. Restate what "These methods" refer to here

To clarify which methods we are referring to we reworded the sentence to read "The methods developed for the measurements in the Uintah Basin GHG network have also been adopted at two UUCON sites to add $CH_4$ observations to the urban $CO_2$ record."

L115: IGRA should be IRGA

We thank the referee for catching this typo. We have fixe it to read the proper acronym IRGA.

L118 - reword - what is continuously flowing (data? air?) and what is high-frequency? not the method.

We agree with the referee that this sentence is a bit confusing and unnecessary. As the next sentence describes that the gasses are continuously flowing and the data is collected at 10-second intervals, we opted to remove this sentence all together.

L123 Again, what exactly is meant by "continuous" vs. "not continuous"? Are the 5 minute points not averages and the 10 second data are averages? Or is this just referring to the higher reporting frequency?

We are grateful to referee for pointing out the fact that we did not define these terms well and that there is confusion surrounding the methods used in the historic data record. A paragraph was added at starting at Line 131 which reads:

"The historic method was a non-continuous method, which collected data on a 5 minute interval. Every 5 minutes a pump would turn on and flow gas for 90 seconds then turn off and the system would then wait 30 seconds for the IRGA to reach a stable pressure. After the stabilization period data was recorded by a datalogger as a 1-minuet average of 10 second scans. The system would then sit idle, with out flowing gasses or recording data until the next sample period."

Additionally, for added clarity, the first sentence on the following paragraph was changed to now read:

"The decision to change from the historical method to that continuously flows gas and collects data was in an effort to better capture higher frequency variations in observed values that could indicate near-field emissions."

L126 I would not put quotes around contamination (not a term being defined here or special usage)

We agree with the referee that it is best to not place quotations around the term contamination in this sentence, and as a result of a comment made by referee #2 we have modified this sentence so that the term contamination has been removed.

The sentence use to read:

"Additionally, high frequency data allow for easier identification of "contamination" of the measurement site from highly localized emissions (e.g., furnace, car) that can affect the signal at a site."

It now reads the following:

"High frequency data allow for easier identification highly localized emissions (e.g., furnace, car) that can affect the signal at a site."

L146. um is not the abbreviation for micrometer, the actual greek mu should be used Here mole fraction should probably be used consistently throughout here rather than switching back and forth with concentration

We would like to thank the referee for catching our error and we have replaced um with the appropriate Greek mu symbol. We also agree with the referee that the term mole fraction should be used consistently, so we have replaced the term concentration with mole fraction here and throughout the rest of the manuscript.

L159 - I was interested in this pump so I looked it up and cannot find this one – can the authors check the number? The NMP850-KNDC (similar part) seems to be a diaphragm pump, not swinging piston. But perhaps the UMP is a piston? Here it would be desirable (as noted at the top of this review) to indicate whether this pump could affect CO2 concentrations downstream.

We are grateful to the referee for their attention to the materials and location of the pump used in the UUCON measurement design, and we agree that the topic requires more information to address the potential for down stream affects. We would also like to thank the reviewer for taking the time to look up the make and model of our pump,

as we had miss identified the model number and the type of pump. The pump is in fact a diaphragm pump and we have corrected the manuscript to reflect that. The sentence on line 181 now reads:

"Atmospheric sample air is pulled from the inlet to the analyzer using a 12-volt chemically resistant micro diaphragm gas pump (UNMP850KNDC-B, KNF Neuberger Inc., Trenton, NJ) that provides a reliable flow of 4.2 L/min."

Additionally, as noted in the response at the top of this review, we have performed multiple tests to verify that absorption and interference are minimal if not negligible and that the materials used were selected with this issue in mind. To address a paragraph was added to this section, section 2.1.3, which reads:

"Since the pump is located upstream of the analyzer there is potential for $CO_2$ to absorb onto the material with in the pump head and interference with the atmospheric sample. The pumps used in the UUCON network were selected to minimize any potential interference with the sample. The diaphragms are made of a PTFE coated EPDM rubber which has been shown to have minimal gas phase absorption. Multiple laboratory and field tests were performed to verify that the location of the pump upstream of the analyzer would not impact the observations. No measureable impacts were identified provide us with a reasonable level of confidence that any absorption or interference from the pump is negligible."

L175 L/minute is volume flow - specify standard liters per minute. I would guess at least some if not all the sites are at altitude, so this becomes more important.

This is an excellent point made by the reviewer and we have update the sentence to note that this is a standard liter per minute. The sentence now reads:

"A Smart-Trek 50 mass-flow controller (Sierra Instruments, Monterey, CA) is located between the manifold and analyzer to hold the sample flow consistent at 0.400 SL/minute (Fig. 4)."
L196 Division's (apostrophe)

We have corrected our error and added the apostrophe.

L200+ I commend the authors on the careful accounting of their standards. It is a little disappointing to hear of drift in the tank of 0.5 ppm however, that seems quite high. (let alone the 1.5 ppm drift in one of the original tanks!).

There appears to be a typo regarding the drift. The range of drift is actually 0.10 to 0.51 $\mu$mol mol-1. We appreciate the referee's comment drawing our attention to this error and the typo has been fixed.

Beyond the typo, drift has been documented by NOAA by measuring a series of cylinders over many years and has attributed most of the drift to water at the valve. The amount of drift that NOAA reports for their cylinders is much smaller than the UofU, please see https://www.esrl.noaa.gov/gmd/ccl/airstandard.html Although the UofU calibration lab's fill system is modeled after the site managed by NOAA, our fill conditions are not the same. NOAA fills their cylinders at an elevation of 3022 meters with clean mountain air while the UofU cylinders are filled at an elevation of about 1288 meters with urban air through an inlet in which both humidity and composition of the air can vary continuously based on human activity. The composition of the air includes species that can interact in the cylinder in unknown ways. We really don't know the type of contaminants that are going into each cylinder that may interfere with the calibration and how the calibration may be affected over time as the cylinder is drained. In addition, we make ideal gas assumptions about gases that are not ideal and can definitely interact at different pressures inside the cylinder as the gas is used up and the intermolecular forces change. All of this amounts to larger drifts in the tanks than those produced by NOAA.

What is the typical value of the target tank?

The target tanks in the network range from 427.79 to 436.84 with an average value of

432.02. To convey this to the reader the following sentence was added on line 229:

"The target tanks were targeted to be slightly elevated above ambient mole fraction, with the average of 432.02 ppm CO2."

L217 subscript 2 in CO2.

We again thank the referee for their careful review of the manuscript and for catching this typo. We have fixed this error and the 2 in CO2 has been subscripted.

L218-221 this paragraph is confusing and a little out of the blue? Is this round-robin between cities occuring or has already been done? (also conduced should be conducted). Or is this a recommendation?

We agree with the referee that this paragraph was a bit confusing and distracted from the detailed information regarding the most recent round robin conducted as described two paragraphs above. For clarity, we opted to remove the paragraph from the manuscript.

L222, In the previous few paragraphs, the units have been \mu mol mol Ë Ę{-1}, now switching to ppm - should be done at the top and defined once and be consistent

We appreciate the reviewer for identifying the inconsistency in the units used in this section. We have updated all instances of umol mol-1 to read ppm.

L226-228. How does this upper limit on the WMO scale affect the calibrations mentioned above which go up to 800 ppm?

This is an excellent question presented by the referee. As the WMO scale was limited on the upper end below the concentrations observed with in the SLV, the facility needed to develop a range of calibration materials that were appropriate to our observational network, while still maintaining tractability to the WMO scale. Thus our calibration range is expanded beyond that of the WMO, but we are still tied directly to the scale. However, the WMO has announced an expansion of the scale, and in response to a

comment from Referee #2, we have added language at line 269 which outlines this expansion. This sentence reads:

"Thus, the current WMO scale may be inadequate for urban observations in the SLV and the announced expansion of the WMO scale to 600 ppm will greatly benefit the urban trace gas community, which needs additional high-quality gas standards with mole fractions more appropriate to urban observations."

L239 $H_2O$ mole fraction (in ppm), and missing "fraction" after "dry mole"

We again thank the referee for their careful reading of the manuscript and for identifying the missing word in this sentence. We have fixed this error and the word fraction has been added to the line.

L254 including Foster et al.? Perhaps a few more words with the ref in parentheses?

We appreciate the referee's suggestion to include more information about how this data has been used in recent publications. To address this comment we have change the original sentence from:

"The Uintah Basin GHG network has supported several recent projects including Foster et al., 2017."

To read:

"The Uintah Basin GHG network has supported several recent projects including Foster et al., 2017 and Foster et al., 2019, in which the data collected from this network were used to estimate and confirm basin wide $CH_4$ emissions and examine $CH_4$ emissions during wintertime stagnation episodes respectively"

The added reference to Foster et al., 2019 has also been added to the References section.

L263 is the NOAA04 scale different from the WMO X2004A scale mentioned earlier for $CH_4$? This is an excellent point made by the referee. Yes, this is the same scale as

described earlier. To clarify the scales as we have reworded the sentence starting on line 238 to read:

"Calibration gases are introduced to the analyzer every three hours using three whole-air, high-pressure reference gas cylinders with known $CO_2$ and $CH_4$ mole fraction that are directly linked to the WMO X2007 $CO_2$ mole fraction scale (Zhao and Tans, 2006) and the WMO X2004A $CH_4$ mole fraction scale (Dlugokencky et al., 2005) as described in section 2.1.6."

L266 section 2.2.2 could be a bit longer - how are these corrections validated in the laboratory? A few more sentences would be useful on this important correction, including an estimate of the error on this (as mentioned below under the uncertainty analysis). At the top of this section (L250) would be good to mention what gases are measured by the LGR.

We thank the referee for encouraging us to include additional information in this section and to further explain our independent validation of the LGR's on-board $H_2O$ corrections. We have added language here briefly describing the lab procedures. Additionally, as the topic of $H_2O$ error estimates is one of significance which was also raised by referee #2, we have produced and included error estimates of the LGR water vapor measurements in this section and in section 4. The sentence now reads:

"The LGR analyzer measures mole fraction of $H_2O$, $CO_2$ and $CH_4$, the later two of which are impacted by the presence of water vapor in the sample and the pressure within the cavity of the instrument. Corrections for pressure, water vapor dilution and spectrum broadening for $CH_4$ and $CO_2$ are made on-site by LGR's software and validated empirically by laboratory testing using calibration gasses of know concentrations and the same Li-610 dew point generator described above, which generates a stable dew point at a set temperature (+/-0.2 °C). Independent error estimates of the LGRs $H_2O$ correction were produced (Section 4, Table 3) resulting in an average uncertainty of 0.017 ppm $CO_2$. "

Figure 5. Caption is incorrect, as it states that the majority of the data is over 550 ppm, which is certainly not the case if the 95th percentile is 550 then only 5% of the data is above 550!

We appreciate the referee for catching this error. We have reworded the caption to now read:

"Figure 5: Monthly percentiles of atmospheric observations from SUG over one year, 2017. Note that observations in the 95th percentile are greater than 550 ppm CO2, well beyond the current WMO calibration scale."

section 3.2: L 288-290 - clarify whether this is an issue in the Licor and LGR both or only Licor. Does the LGR control cavity pressure? Would the constantly flowing reference gas in the case of the Licor take care of this issue? (perhaps not).

We are grateful to the referee's suggestion to clarify what method of accounting for changes in atmospheric pressure each network utilizes. We have added language to section 3.2 that clarifies this. This new language reads:

"To account for pressure the LGR's control the pressure with in the cavity and maintaining a near constant 140 torr. The Li-6262's in the UUCON network do not have mechanisms for controlling the pressure with in the cavity and thus implement the latter strategy described above, calibrating frequently and standardizing the flow of gasses through the optical cavity"

L303: check consistency between British and American spelling of "vapor/vapour"

All instances of the spelling vapor has been standardized to the American spelling.

L305: is this now a mole fraction of H2O or something else (volume fraction, mass fraction? Mixing ratio often refers to mass ratio?)

We appreciate the referee catching this error. This is H2O mole fraction.

The word "mixing ratio" was replaced with "mole fraction" on line 410.

Section 3.3 Please remind the reader that this is only being done for UUCON because the LGR is doing its own correction interally.

Language was added to both the opening paragraph and the final sentence of section 3.3 to remind the reader that this is only applied to the UUCON sites. That language reads:

"Both of these effects are corrected for during the post processing of UUCON data while the LGR sites rely on LGR's internal software."

And the last sentence was modified to end saying:

"mole fraction with in the UUCON network."

Fig. 7 caption wording: comma should be semicolon? (before flush periods). And here the target is referred to as the check tank, it might be more consistent to refer to it here as the target tank as was done earlier. Also, should say "calibrated", not post calibrated. Or perhaps "post-calibration".

We thank the referee for the careful review of the figure caption and identifying these errors. Each error has been corrected and the figure caption now reads:

"Figure 7: Left panel shows the sequence and timing of a standard calibration period in both the UUCON and Uinta Basin network. Gray open circles indicate the 90 second flushing period observed between each change in gas. Right panel shows a full two hour sample period with calibrations for the UUCON network with linear interpolations; flush periods have been removed. Orange, green, and blue closed circles indicate calibration standard gas and their known $CO_2$ concentration. Yellow closed circle represents the target tank and its known concentration. Black closed circles indicate pre-calibration atmospheric observations which have been down sampled to one minute averages to reduce over plotting. Plus (+) signs in all colors indicate the calibrated measurements for the corresponding measurement. "

L359: It has not been shown what typical uncertainty due to these might be for this

system. Perhaps the authors could indicate what typical analyzer precision is during calibrations for the two instruments (I now see this information is already in Table 1, so you could just refer to this here). More difficult is the water vapor correction – are there any thoughts as to the possible error here, even if just a guess or based on a few tests mentioned earlier where the LGR correction was checked in the lab – what do those tests show? It would be good to be able to back this statement up, that these are likely small compared with the running uncertainty, even if just with some anecdotal information, without a formal estimate of their values.

We thank the referee again for pointing out the importance of including uncertainty estimates for the water vapor for both systems. As mentioned above, we have calculated these estimates for each system and reported on them in table 3 along with the CO2 measurement precision mentioned by the referee. We described the methods for these calculations in the added language in section 2.2.2 lines 338 – 364. To address the referees specific comment here of providing information to back up the statement that the propagated errors of H2O uncertainty are small we removed the words water vapor from line 438, added a sentence at the end of the opening paragraph of section 4 which reads:

"Due to the importance of water vapor on the accurate measurement of a CO2, especially in a measurement system that does not dry the atmospheric sample like the two describe in this paper, we have produced and reported uncertainty estimates for H2O vapor measurements ($1\sigma$ Uh2o) as it impacts COǑ2 as well as observed analyzer precision ($1\sigma$ Up) in the field (Table 3). We do not report a total, accumulative uncertainty estimate from all possible sources of error combined. Uncertainties beyond those reported here are small compared to the running uncertainty estimate and could be estimated in future work."

We have described the methods used to create this statistical uncertainty estimate in section 4, lines 542 - 553, which reads:

"Water vapor precision was examined using laboratory tests for the UUCON and the Uintah Basin GHG network designs and are reported in Table 3 (Uh2o). Gas from a dry calibration tank of know $CO_2$ mole fraction was passed through a Li-610 dew point generator at a set dewpoint temperature. $H_2O$ measurements were collected by both systems in parallel over a period of 1.5 hours. We calculated the Allan variance to represent the precision of the $H_2O$ measurements regardless of drift over time or other systematic errors. This precision statistic was used to construct a normal distribution of $H_2O$ centered on the mean measured $H_2O$ mole fraction at each site, which is used to estimate the uncertainties in dry air equivalent estimates for $CO_2$ due to $H_2O$ repeatability error using methods discussed in Section 3.3. The $1\sigma$ uncertainty of the $H_2O$ precision results in a mean 0.019 ppm $CO_2$ error (Uh2o) for the UUCON network, and 0.017 ppm $CO_2$ for the Uintah Basin GHG network design. These uncertainties represent a lower bounds for error in $CO_2$ resulting in $H_2O$ measurements as they do not account for errors in $H_2O$ measurement accuracy. Biases in the accuracy of measurements are addressed in the QAQC procedures of the data."

L361: clarify that this is only available at the UUCON stations

A sentence was added on line 480 that reads:

"Since the UUCON network design encompasses a target tank we are able to leverage this method to estimate uncertainty within the network"

L401. I would think this method would actually not be able to see a bias in the calibration value of a given tank, it only can see whether the calibrations are noisy from day to day, because you are interpolating across calibration periods. For example, if on tank is always biased high due to a bias in the assigned value by the calibration lab, the slope has a bias all the time, and this would not be included in the virtual target method. Unless I am not understanding this right. If I am then it would be good to mention a case where this method does not capture the true full uncertainty that a true target would, or at least provide a sentence as to the limitations. Otherwise, this seems

like a good method.

We thank the referee for the comment regarding the methods ability to account for tank bias, but we respectfully disagree with this comment. For the interpolated target tank method describe (UpTGT) .The interpolation between the t-1 and t+1 calibrations allows us to evaluate each calibration sequence individually at time t, and also to provide an uncertainty calculation for each calibration at time t. The smoothing we applied to the RMSE values reduces the variability, but it does not bias the uncertainty values.

Thus, we feel this method can detect a bias in a calibration tank because any bias would result in larger measured-known calibration tank values.

For example, using the reviewer's scenario, if one tank was inadvertently biased high due to a bias in the assigned value by the calibration lab, the slope would be biased also and this would result in a higher virtual target tank RMSE values while this tank was used. Something like this is probably the explanation for why the June to early Aug RMSE values were elevated as compared to the latter time periods seen in Fig 9.

L408 typo: sites

This typo has been corrected and we reworded the sentence to now read:

L 409 - Is there an overall difference in the mean? i.e. Is UTGT larger than UpTGT on average? I would think so, but it would be nice to state the mean difference as well as the mean absolute difference here. I now see this in table 1 where it seems that at most sites where both were computed UTGT is greater, but not at all. this could be mentioned here, with a reference to the table.

We appreciate the referee's suggestion to reference table 3 with in the final paragraph of section 4. To address this comment we gave added language in line 527 which reads ", with average values reported in Table 3." As well as citing table 3 at the end of the section on line 541.

Table 3: are these the mean uncertainties over some period of time? Please note in

the caption.

Yes, these numbers are the average uncertainty, precision and data recovery rates for the entire data record. We have updated the caption to note this. It now reads:

"Table 3: $CO_2$ and $CH_4$ Measurement Uncertainties with Gaussian window target tank method (UpTGT), target tank (UTGT), analyzer precision at $1\sigma$ (UP), $H_2O$ measurement precision $1\sigma$ (Uh2o) as expressed in ppm $CO_2$ uncertainty, and data recovery rates from UUCON and Uintah Basin GHG measurement averaged over the entire record since sites were overhauled."

L416: calibrated rather than post calibrated?

We agree with the authors suggested change and have updated the sentence to read calibrated.

L424: ppm should be ppb for $CH_4$ here.

We thank the author for catching this error and we have changed the units from ppm to ppb.

L435. Is this proper ordering of sections? Data Availability before Conclusions

Yes, ESSD guidelines do specify that the data availability section comes before Conclusions.

L454. Why does the UpTGT method likely overestimate uncertainty? There is no other indication of this anywhere in the text.

This is an excellent point made by the referee, and one that referee #2 also made. We agree that the evidence suggests that this method is highly similar to other uncertainty metrics. Thus we have removed this sentence all together. The sentence use to read:

"While this method likely results in overestimation in the uncertainty, this novel method for estimating uncertainty nonetheless provides useful insight into the quality of data

produced at individual sites and is broadly applicable to any atmospheric trace gas or air quality dataset that contains calibration information."

The sentence now reads:

"This novel method for estimating uncertainty provides useful insight into the quality of data produced at individual sites and is broadly applicable to any atmospheric trace gas or air quality dataset that contains calibration information."

Anonymous Referee #2

This manuscript presents the data from the UUCON and Uintah basin GHG networks. The dataset is a valuable addition to the urban GHG research field, and the manuscript overall is well-written in describing the instrumentation and uncertainty analysis associated with this dataset.

I suggest minor revisions to address a few notable questions, namely:

– I feel that one of the most difficult aspects of an urban GHG monitoring network is the delicate balance between being in close to the emissions in the area but not "too close", as in being influenced by emissions in very close proximity to the measurement site that would be difficult to interpret from a modeling perspective. The authors actually mention this "contamination" in Ln126, as one of the motivations to implement changes to the instrumentation that would better capture high-frequency variations. So, have the authors looked into whether local "contamination" can be identified in the dataset? I feel a flag that identifies periods of potential local "contamination" could be of great value to the potential user of the data, as such screening may require detailed knowledge of the site environement that only the authors could have. If not for this paper, perhaps another study could be done that delves deeper into this issue?

We thank the referee for making this important point, and agree that one of the difficulties in near surface urban measurements is in understanding how near field emissions can impact the signal at any given site. We feel that the use of the word "contamination"

may have been unclear as to our point, and in response to a comment by referee #1, we have replaced the term contamination with "near field emissions" on line for clarity.

We appreciate the referee's suggestion of a flag with in the dataset that identifies periods of potential near field emission signatures. While we agree that this sort of flag could be useful, we feel that this is more of an analysis decision to be made by the end user. Any metric by which we calculate and flag the data may not fit the needs of each user, and thus we leave this sort of analysis up to the end-user.

To the referee's larger point of how to best utilize a dataset that captures near field emissions and larger scale urban emissions, while we feel that this subject is beyond the scope of this paper, recent work by individuals with in our group, and co-authors on this manuscript, have published a paper using WRF-HRRR to characterize near field emissions with a similar dataset from the Salt Lake Valley (Fasoli et al., 2018; Simulating atmospheric tracer concentrations for spatially distributed receptors: updates to the Stochastic Time-Inverted Lagrangian Transport model's R interface). Thus the near field emission captured in datasets like this one can be a useful signal to capture and the modeling framework to understand these signals is under development and ongoing.

– I find it surprising that the Uptgt results presented in this study seem quite high compared to those from other networks, and also that Figures 10, 11 suggest signficiant variation and trends in this value. The authors do mention differences in instrumentation compared to other networks and influence from the environement where the instruments are operated in as possible explanations for these results, but I do wonder if a more thorough investigation is warranted to really understand this issue.

We appreciate the referee for pointing out some of the variability with in the uncertainty estimates displayed in figures 10 and 11. This variability can be driven by several very important changes at the sites that were not originally discussed in the paper. These include the fact that any bias in the assigned calibration tank values will result in a

sustained increase in the uncertainty metric, and that the distribution of concentrations over the span range can impact this metric. Thus by changing one or multiple tanks at the site we can end up with step wise changes in the calculated uncertainty metric. Lastly, the metric UpTGT described in this paper is impacted largely by the time between consecutive calibration periods, thus periods of missing data can result in high uncertainty before and after the data gaps. The suggestion from the reviewer to further investigate these features of UpTGT was a critical suggestion, as we examined the time dependence of the metric further. This resulted in us implementing an 8 hour mask, in which we remove any period from the analysis where there is an 8 or more hour gap in the data. This has slightly improved our results and reduced some of the larger variability seen in Figures 10 and 11.

To address this, language has been added in section 4.0,line 534, that describes each of these factors. This language reads

"It should be noted that since UpTGT is time dependent, gaps in data will result in large uncertainties estimates. As a result we have added a mask, in which any period of 8 hours or more of data are removed from the UpTGT calculation. Additionally, bias in the assigned values of calibration tanks, as well as changes in the distribution of the mole fraction of calibration tanks on site, can result in result in step wise changes in UpTGT as can be seen if figures 10 and 11."

Additionally, of equal importance to the overall improvement of the manuscript, during our further investigation of UpTGT we discovered an error in our original reporting of the UTGT metric. In Table 3, and the numbers described in the text, we mistakenly reported the native values of UTGT and not the convoluted values for UTGT. As UpTGT is an 11 hour Gaussian convolution, we want UTGT to best match. We have updated the numbers in Table 3 and have update the values throughout the text to correct this error. The correctly reported numbers result in a significantly better match between UpTGT and UTGT, further emphasizing the power of the novel method.

– Because the calbration/target tanks are dry and the air measurements are made "wet" (without any drying), it seems like uncertainties in the moisture correction would become a significant source of uncertainty in the air measurements, and this uncertainty would not be captured in Uptgt, Utgt. Do the authors have thoughts on this issue? Should some representation of the moisture correction uncertainty be presented to the users of the data?

We would like thank the referee for pointing out this very important potential source of error with in our measurement design, and note that referee #1 also commented on the errors associated with water vapor corrections. To address this comment, we have estimated the measurement uncertainty of the water vapor measurements for both the UUCON and the Uintah Basin GHG networks and report those values in Section 4 and in Table 3. The measurement errors have been propagated into $CO_2$ error by applying these errors to the water vapor band broadening and dilution corrections. The most significant added language on line 543 now reads:

"Water vapor precision was examined using laboratory tests for the UUCON and the Uintah Basin GHG network designs and are reported in Table 3 (Uh2o). Gas from a dry calibration tank of know $CO_2$ mole fraction was passed through a Li-610 dew point generator at a set dewpoint temperature. $H_2O$ measurements were collected by both systems in parallel over a period of 1.5 hours. We calculated the Allan variance to represent the precision of the $H_2O$ measurements regardless of drift over time or other systematic errors. This precision statistic was used to construct a normal distribution of $H_2O$ centered on the mean measured $H_2O$ mole fraction at each site, which is used to estimate the uncertainties in dry air equivalent estimates for $CO_2$ due to $H_2O$ repeatability error using methods discussed in Section 3.3. The $1\sigma$ uncertainty of the $H_2O$ precision results in a mean 0.019 ppm $CO_2$ error (Uh2o) for the UUCON network, and 0.017 ppm $CO_2$ for the Uintah Basin GHG network design. These uncertainties represent a lower bounds for error in $CO_2$ resulting in $H_2O$ measurements as they do not account for errors in $H_2O$ measurement accuracy. Biases in the accuracy of

measurements are addressed in the QAQC procedures of the data."

– I'm a bit puzzled by the relatively poor performance of the LGR instruments compared to the older Licor instruments. Just out of curiosity, I downloaded the FRU station data (the worst instrument in terms of precision), looked at the numbers for a handful of individual cal runs, and the stdev's I see seem much closer to those of the Licor instruments than to the 1-sigma precisions for the LGR instruments reported in Table 3. I would like to ask that the authors check their calculations and make sure they can stand by the results and discussions presented in this study.

We appreciate the referee's attention to the precision calculations reported in table 3, and are happy to hear that they have taken advantage of our online platform to look at the data in detail. We have double-checked our calculations and have produced the same numbers. Additional lab tests were performed using calibration gasses of know mole fraction, which produced comparable numbers. One note of importance, the LGR data sheets report an Allan variance as their standard deviation while we are reporting a single standard deviation.

Minor comments below:

Ln23: Remove space in the doi link, e.g. ".org/ 10." -> ".org/10."

We thank the referee for catching this typo and we have resolve it within the text. The DOI link now reads:

"https://doi.org/10.7289/V50R9MN2

Ln37: I'm not sure that stating a flask network is "expensive to operate" is a sound scientific statement, I think most would agree that there are substantial costs to any type of measurement effort, and one might even argue that a flask network is a more affordable way to operate a dense, wide measurement network, if you are willing to sacrifice measurement frequency. I would suggest that the statement be removed.

The referee makes an excellent point regarding the substantial costs of any measurement effort. We agree with the referee's suggestion to remove the statement and we have deleted it from the manuscript. This sentence use to read:

"Flask-based sampling networks such as the one led by NOAA-Earth System Research Laboratory (Tans & Conway 2005; Turnbull et al., 2012) offer long-term, globally representative records of several atmospheric tracers but can be expensive to operate, create temporally sparse datasets, and often do not capture intra-city signals."

It now reads:

"Flask-based sampling networks such as the one led by NOAA-Earth System Research Laboratory (Tans & Conway 2005; Turnbull et al., 2012) offer long-term, globally representative records of several atmospheric tracers, however their measurement frequency is generally limited, and often do not capture intra-city signals."

Ln105: Authors should clarify the "These methods" referenced here.

We thank the referee for asking us to clarify this comment, and note that referee #1 made the same comment. To clarify which methods we are referring to we reworded the sentence to read "The methods developed for the measurements in the Uintah Basin GHG network have also been adopted at two UUCON sites to add CH4 observations to the urban CO2 record."

Ln115: IGRA -> IRGA

This typo has been resolved.

Ln123: The change from a non-continuous 5-min collection to the 10-sec data collection requires a bit more explanation. My guess is that the 5L mixing buffer mentioned in Ln139 is essentially what the authors refer to here, with the assumption that the residence (mixing) time in the buffer is about 5 minutes? Perhaps the paragraphs here can be rearranged to make this point more coherent?

We are grateful for the referee pushing us to clarify the language in this paragraph, and

again note that Referee #1 made this same comment, stressing the unclear nature of the original wording. To clarify this section, a paragraph was added at starting at Line 131, which reads:

"The historic method was a non-continuous method, which collected data on a 5 minute interval. Every 5 minutes a pump would turn on and flow gas for 90 seconds then turn off and the system would then wait 30 seconds for the IRGA to reach a stable pressure. After the stabilization period data was recorded by a datalogger as a 1-minuet average of 10 second scans. The system would then sit idle, with out flowing gasses or recording data until the next sample period."

Additionally, for added clarity, the first sentence on the following paragraph was changed to now read:

"The decision to change from the historical method to that continuously flows gas and collects data was in an effort to better capture higher frequency variations in observed values that could indicate near-field emissions."

Ln159: Do the authors mean "NMP" instead of "UMP"? Can the authors clarify the diaphragm and pump head materials for the pump?

We thank the referee for catching this error on our part and we have resolved the typo and changed the part number from NMP to UMP. At the suggestion of Referee #1, have also added substantial language around the pump materials and testing done to validate the lack of absorption / interference of the pump with the atmospheric sample. This language was added on line 181, which reads:

"Atmospheric sample air is pulled from the inlet to the analyzer using a 12-volt chemically resistant micro diaphragm gas pump (UNMP850KNDC-B, KNF Neuberger Inc., Trenton, NJ) that provides a reliable flow of 4.2 L/min."

Additionally a paragraph was added to this section, section 2.1.3 line 188, which reads:

"Since the pump is located upstream of the analyzer there is potential for $CO_2$ to absorb

onto the material with in the pump head and interference with the atmospheric sample. The pumps used in the UUCON network were selected to minimize any potential interference with the sample. The diaphragms are made of a PTFE coated EPDM rubber which has been shown to have minimal gas phase absorption. Multiple laboratory and field tests were performed to verify that the location of the pump upstream of the analyzer would not impact the observations. No measureable impacts were identified provide us with a reasonable level of confidence that any absorption or interference from the pump is negligible."

Ln162: Bev-A-Line OD/ID's?

The Bev-A-Line is $\frac{1}{4}$" OD and 1/8" ID. To convey this information to the reader, language was added to line 185 that reads "This loop is comprised of at least 9 meters of $\frac{1}{4}$" outer diameter (OD) (1/8" inner diameter) Bev-A-Line to provide sufficient resistance to the gas so when the manifold is open, gas passes through the mass flow controller and into the analyzer at the desired rate without losing all of the gas to the sample loop bypass (Fig. 4)."

Ln167: What is the material and approx. inner volume of the gas manifold? I assume this was addressed by the 90 sec flushing time, but did the authors worry about dead volumes in the manifold? Ditto for the LGR Multiport Input Unit mentioned in Ln263?

The manifold is made out of anodized aluminum and has an approximate diameter of 0.004 L. Yes, dead head was a concern and has been addressed both by the 90 second flush period, as well as the ordering of the valves, with the atmospheric sample valve always located at the back of the manifold allowing for a full flush of any potential dead headed gasses. The MIU for the LGR's are made from a stainless steel and have a similarly small internal volume of $\sim$0.008 L.

Ln179: How long do the calibration tanks last out in the field? What model regulators used?

Generally, baring a leak, calibration tanks last at least 12 months in the field. We have added a sentence on line 216 to include this information. This sentence now reads:

"Each site houses three whole-air, high-pressure cylinders with known CO2 mole fraction which are directly linked to World Meteorological Organization X2007 CO2 mole fraction scale (Zhao and Tans, 2006), which generally last around one year in the field."

Because the network dates back more than 15 years there is a wide array of regulators used, and providing the model numbers of all these would be beyond the scope of this manuscript. However, when regulators are purchased for the network now we tend to use VWR, brass, single stage regulators with either neoprene or stainless steel diaphragms (like model 55850-620).

Ln182: Perhaps "ID -99" can be removed here, as it is a bit confusing without context, and repeated in Ln 328.

We agree with the referee that the reference to the ID -99 is with out context here and has been removed.

Ln194: Can the authors clarify the "150L" capacity of the aluminum tanks? My guess is that the authors refer to the N150 style tanks, with 29.5L internal volume.

The referee is correct; the gas cylinders are classified as N150 aluminum cylinders with a volume of 29.5 liters. We have modified the text to correct this error. It now reads:

"29.5 L volume N150 CGA-590 aluminum tanks are filled with city air using a high-pressure oil free industrial compressor (SA-3 and SA-6, RIX Industries, Benicia, CA)."

Ln198: How was the âĹij5000 ppm CO2 spike tank sourced? Is spiking for CH4 done separately using an additional tank?

We appreciate the referee's detailed knowledge and question surrounding the sourcing and production of our calibration materials. The high spike CO2 tank is filled in the calibration lab by taking an aliquot from a 100% CO2 gas cylinder and filling it with

dried SLC air. The CH4 high spike tank is prepared in the same manner from a 998 ppm CH4 cylinder purchased from Airgas. Two separate aliquots from each cylinder type created for the targeted gas fill are used when filling a cylinder that has a target of both CO2 and CH4. To convey this information to the reader language was added on line 237 which reads:

"This spike tank was filled in the calibration lab by taking an aliquot from a 100% CO2 gas cylinder and filling it with dried atmospheric air."

And on line 262, which reads:

"The spike tank used to produce elevated CH4 calibration tanks was generated using the same method as the CO2 spike tank, but using an aliquot from a 998 ppm CH4 cylinder purchased from Airgas, Inc (Pennsylvania) and filling it with dried atmospheric air."

Ln199: Any special air conditions in which the compressors are operated? Any need for dilution to bring the concentrations down in the tanks, and if so how is this done?

We again thank the referee for their focus on the production of our calibration materials, and appreciate the improvements to the manuscript gained by these comments.

There are no special air conditions that are required to run the air compressor. However, when CO2 and CH4 are excessively high due to human activity near the inlet, the timing of the fills need to be considered based on the target.

To fill cylinders below ambient levels of CO2 a diluent is made in the calibration lab. To create the diluent for ambient CO2, a soda lime trap is used to scrub the CO2 from the ambient air however the reaction produces water which requires additional magnesium perchlorate to dry the scrubbed air. The soda lime trap is set up inline first, followed by two magnesium perchlorate traps to ensure that the scrubbed air fill contains only dry air. The calibration lab is unable to fill a diluent gas cylinder for CH4 dilution fills, therefore zero air is purchased from Airgas for this purpose. However, fills

of sub-ambient CH4 tanks are rarely done.

To convey this information to the reader, language was added on line 238, which reads:

"To produce sub-ambient calibration tanks, tanks are mixed with a diluent made from atmospheric air scrubbed with a soda lime and magnesium perchlorate trap."

Ln 203: Cause of assignment difference on the tanks? Misassignment or tank drift? Has previous data be updated to reflect the assignment change in the lab standards?

A typo in this section was corrected with substantially reduced the reported drift from 0.10 to 1.52 ppm to 0.10 to 0.51 ppm. Additionally, as addressed above when responding to a similar comment from Referee #1, the larger drift noticed in the UofU tanks compared to NOAA tanks can be attributed to the fact that UofU tanks are filled in an urban area and at a substantially different elevation.

Ln216: Have the authors thought about calibrating CO2 on the LGR as well?

We thank the reviewer for this suggestion. Yes, When we began using the LGR for CH4 calibrations, we ran a series of comparisons to determine which instrument had the best reproducibility of CO2. At the time the Licor instrument was determined to provide the most reproducible measurements of CO2.

Ln217: Subscript missing in "CO2"

We thank the referee for catching this error and it has been resolved within the text.

Ln225: The authors should note that NOAA is actually working on extending the CO2 scale to 600 ppm, see presentation from the last GGMT meeting in Switzerland: https://www.wmo.int/pages/prog/arep/gaw/documents/GGMT2017_T04_Hall.pdf

We appreciate the referee pointing us to the GGMT presentation describing the efforts to expand the WMO scale to 600 ppm. To note this, we have change the language from:

"Thus, the current WMO scale may be inadequate for urban observations in the SLV. The urban trace gas community should consider developing and sharing additional high-quality gas standards with mole fractions more appropriate to urban observations."

To:

"Thus, the current WMO scale may be inadequate for urban observations in the SLV and the announced expansion of the WMO scale to 600 ppm will greatly benefit the urban trace gas community , which needs additional high-quality gas standards"

Ln241: Description of the internet stream that provides the network clock is missing. I presume this same internet connection is also used to send the data to the Univ of Utah data server?

We thank the referee for pointing out the lack of description of our internet connections at the measurement sites. Language was added at line 315 that briefly describes our internet hardware, which reads:

"Network time checks and data transfers are established via internet connections at each site either through existing ethernet connections or cellular modems (RV50, Sierra Wireless, Carlsbad, CA)."

LN250: I'm not sure exactly what the authors refer to with "off the shelf". I presume what the authors mean is that the LGR's were largely operated using peripherials from LGR and corrected using LGR's internal algorithms, such that the complete system was "off the shelf". However, I don't think it's fair to characterize the LGR anlyzer itself as more "off the shelf" than the Licor 6262's, I'm sure there are "off the shelf" ways to operate the Licor's as well. I suggest the authors clean up the message here.

We appreciate the referee's suggestion to clean up the language around our "off the shelf" messaging. Language was added at line 323 to clarify our message, which reads: "Unlike the UUCON network, in which the measurement system and it's peripheries are essentially a custom engineered solution of an array of different components

from multiple manufactures brought together by the researchers running the network, the LGR sites employ systems fully designed by a single manufacture. The use of an "off the shelf" unit like that deployed in the Uintah Basing GHG network has both advantages and disadvantages."

Also, can you clarify whether the LGR data is calibrated from the internal software, or instead worked up in post processing like the Licor data?

We thank the referee for asking for clarity around the calibration of the data from the Uintah Basin GHG analyzer. We specify on line 326 that the same post processing calibrations scripts are used in this network, For added clarity, we added the term post processing so the sentence now reads:

"In an effort to minimize differences between the two networks, measurement frequency, networking, calibration materials (sections 2.1.6), and post processing calibration methods (section 3.1) all follow the same protocols described for the UUCON network with the notable exception of the calibration frequency, which is every three hours as opposed to every two with the Li-6262's."

Ln260: Just to make sure, the same protocol of 3 cal gasess that range the actual atmospheric data, prepared in-house?

That is correct. To add clarity a sentence was added at line 341 that reads, "Molar fraction of CH4 calibration gasses are chosen to align with the 5th, 50th and 95th percentile of the previous years observations, while CO2 gasses match those described in section 2.1.6. "

Ln273: Reword sentence to simply say it wasn't implemented.

To address this comment the sentence was reworded from:

"However, the current version of the LGR proprietary software that drives the MIU calibration unit lacks flexibility to accommodate a calibration sequence independent of a standard sequence. Thus, the off the shelf nature makes the implementation of this

somewhat more difficult."

To:

"However, the current version of the LGR proprietary software that drives the MIU calibration unit lacks flexibility to accommodate a calibration sequence independent of a standard sequence and thus a target tank was not implemented in the Unitah Basin GHG network design."

Ln274: Can the authors clarify how much of the description in Section 3 apply both to the Li-6262's and the Lgr's, as opposed to only the Li-6262's?

We thank the author for noting the vague nature of Section 3, leaving the reader confused as to which network this section applies to. We also note that referee #1 made the same comment. To clarify, language was added to the opening paragraph of section 3 which reads:

"For both the UUCON and the Uintah Basin GGA network, raw"

Ln336: Can the authors discuss what checks are implemented in the "automated quality control scripts"?

Absolutely. We would like to note that currently we set the tolerances for automatic removal of data to be very wide. Thus much of the data deemed bad and removed is selected by qualified technicians at the University who can best identify why data is bad and determine if it can be salvaged by some minor changes or if it needs to be fully scrubbed from the published data. The QC checks currently used in the UUCON and Uintah Basing GHG networks include a verity of parameters including:

If there is an absolute difference greater than 10ppm $CO_2$ between the analog and the serial measurements in the UUCON network, the serial is replaced with the analog reading. This was the result of issues with the serial buffer on some of the Li-6262's, which has been resolved in the later part of the record. If the reading from the mass flow controller is outside of the range of 395 – 405 Ml/min we flag the data but still include

it in the dataset. If the technician failed to input the known standard concentrations at any site it is flagged but included in the data. Flushing periods are removed. Periods identified by a user as problematic are removed. If the time between calibrations is greater than 5 hours that data is removed, if the calibration tank is measured more than 100 ppm off from the known value it is removed. Lastly, for the LGR sites, if the cavity pressure is outside of the range of 135 – 145 torr it is flagged.

Lastly, bias in the H2O measurements are identified by comparisons to other near by H2O observations and corrections are applied by a qualified technician to account for any identified basis. There are a wide array of options for this process which will be discussed in-depth in a paper that is currently in preparation that describes the historic network that predates the overhaul described in this paper.

We have chosen not to include these specifics in the manuscript since the specific tolerances used currently can be changed and new QC protocols can be added at any time as new problems are identified in the network. Thus by pointing to the online repository where these details are pulled from allows for the most up to date information to be accessible by the reader.

Ln374: 22 hours, according to Verhulst et al. 2017

We thank the referee for catching this error, and we have corrected the text to read "every 22 hours"

Ln395: I don't see blue circles in Figure 8d, but I do see gray circles with a blue line going through them. . .

We appreciate the referee's comment noting that the color selection in this figure was difficult to discern. We have updated the figure so that the blue circles that appeared gray are now blue squares and have updated the text to represent this change.

Ln400: I suggest that the explanation of the yellow shaded region on this line be re-peated on the captions for figure 9.

We are grateful to the referee for this suggestion as it adds significant clarity to the figure. We have added the following language to the caption:

"The yellow shaded region in Figure 9 is the time period shown in Figure 8 "

Ln401: "calibrated target tank mole fractions", I initially understood that to mean the concentrations assigned to the target tank from the central lab. Would it be a bit more clear to say "on-site assignment of the target tank"?

We thank the referee for pointing out the confusing nature of this statement, and we have modified line 520 in the track changers document it to read:

"In July-August 2017 at IMC there was a bias in the post-calibration target tank mole fractions that similarly affected the pseudo target tank RMSE values (Fig. 9d)."

By changing it to read "post-calibration", this sentence now better matches the nomenclature used elsewhere in the manuscript to describe the values after calibrations.

Ln402: Can you clarify the "third calibration tank"? Was it the tank with the highest concentration?

We again thank the referee for pushing us to clarify some of the language with in this section and associated figures. The lowest calibration tank was removed during this period. We have added "low concentration calibration tank" to line 521 to add this clarity.

Ln402: Not sure why Figure 8d is referenced here?

We appreciate the referee catching this error, as figure 9d was the appropriate reference. We have corrected the text to fix this error.

Ln404: After the "third calibration tank" is re-installed, there is a brief jump in TGT RMSE. Is there an explanation for this? Also, there's a gap in the pTGT calculations at the same time, is there an explanation for this?

We thank the referee for taking the time to carefully examine our figures and take note of the elevated pTGT values overved in Figure 9. After careful examination, we discovered that the data missing in that gap was flagged and removed by a technician, however they failed to remove the calibration tank data along with the atmospheric and calibration data. As a result, the target tank data that drove the elevated rmse calculation in this figure should not have been included. We have remove the target tank data from the dataset and reproduced the figure, which no longer shows the elevated rmse pTGT values.

Ln407: Are the hourly datafiles with the Up-TGT estimates considered an additional data level, compared to discussions in section 3.4?

No, these files are solely produced for the calculation of UpTGT and are not considered a data level. If end users are interested in hourly datasets the data is formatted in a way that it is easy to produce averaged files.

Ln429-430: This is an interesting observation. I think one important distinction to make here is that the two analyzers were operated with different peripherals, such that the comparison isn't completely apples-to-apples, per se. I'm also somewhat concerned that the LGR precisions are significantly worse than those suggested in LGR data sheets (100 sec 1-sigma precision for CO2 0.05 ppm, CH4 0.3 ppb). While I under stand that company data sheets are not to be trusted, I'm surprised that the LGR's show worse precision than the 20 year old Licor's! Have you tried directly replacing the Licor analyzer with the LGR? I understand that the temperature of the lab can have a significant effect on instrument performance, but I would have suspected that most of those effects would concern instrument drift, and that short-term precisions are relatively less affected, so I find the high Up values for the LGR surprising. Can the authors get into a bit more detail on what might be causing these findings?

We thank the referee for their interest in the discrepancy in the values we observed for the LGR precision compared to the datasheets. We were also surprised at these

findings and worked with Los Gatos to identify the source of the differences. There is one key factor that contributes to this discrepancy. First, the LGR datasheets are reporting an Allan Standard Deviation (Allan variance) at 100 seconds while we are reporting a true 1-sigma standard deviation around 10 second observations

Ln454: One thing that's clear is that Uptgt is generally higher than Utgt, at least based on eyeballing Figure 10. However, can you be sure that Utgt doesn't underestimate the data uncertainty, due to the fact that the 25hr sampling sequence doesn't fully capture the calibration uncertainties that happen at 3âĹij6hr intervals?

We appreciate the referee's point regarding Utgt's potential to underestimate uncertainty as a result of the difference between the 25 hour sampling sequence of the target tank vs lower frequency variability in the measurements. While we agree that this is a possibility, if there was variability at 3-6hrs it would manifest itself in our 25-hr target tank sample frequency as apparent noise. However we do not see noise in the target tank observations, and are thus confident in Utgt methods ability to accurately predict measurement uncertainty.

Please check subscripts for CO2, CH4 in the reference list.

We again thank the referee for their attention to details and catching the missing subscripts throughout the reference section. We have searched the references and subscripted where appropriate.

Multiple Mitchell et al. 2018 references on list, should clarify in accordance with ESSD style guidelines.

We again thank the referee for catching the error in the reference list and have corrected the order of the Mitchell et al., 2018 citations to match ESSD style guidelines.

Table 1: Lat/Lon's should specify N, W.

N and W have been added to the header in table 1 in the latitude and longitude columns.

Figure 4: Can the authors clarify the tubing used in the gas connections in the main text?

Unfortunately there is not a fully standardized use of the two types of tubing materials utilized in the UUCON network, and thus there are very subtle differences between sites. Additionally, there are often points where the two materials are interconnected for short distances, such as barbed connections to filters and the manifold, that complicate that tubing selections beyond what is relevant to the reader. Thus we feel that a substantial amount of text would be required to provide this level of detail to the reader, which wouldn't greatly improve their understanding of the network design.

Figure 7: "Check gas"? Why not "target" tank, just to be consistent?

We thank the author for identifying this error. The caption has been updated from check gas to target tank.

Figure 9. I'm not sure what 9B is showing here. Also, is there some gray in the background of 9D, and what does that reflect?

Figure 9B displays the same information as 8B but over a significantly longer time frame and thus the small scale variability is not noticeable. However the changes in calibration tanks is very noticeable as the high concentration tank drops from 570 to 480 ppm, and there is a period in the low concentration tanks that forces changes in the calibrated TGT values in panel C and the UpTGT and UTGT values in panel D. Unfortunately this is too much information to include in a figure caption. To add some clarity, the language "See description in text (section 4) for greater details." The gray in the figure represents UpTGT prior to the 11point convolution. To add clarity to this, we have darkened the color so it appears better in the legend and the figure.

Figures 10-11. I'm surprised that there is so much variation in Uptgt, especially for CH4 where it seems like there are clear jumps (likely at cal tank changes) and long-term drift. Have the authors looked into the causce of these uncertainties? For example I wonder

if the spikes in RMSE errors correlate with large spikes in the air concentrations, in which case one may suspect a leak or memory effect in the system?

We are grateful to the referee for pointing out some of the variability with in the uncertainty estimates displayed in figures 10 and 11. This variability can be driven by several very important changes at the sites that were not originally discussed in the paper. These include the fact that any bias in the assigned calibration tank values will result in a sustained increase in the uncertainty metric, and that the distribution of concentrations over the span range can impact this metric. Thus by changing one or multiple tanks at the site we can end up with step wise changes in the calculated uncertainty metric. Lastly, the metric UpTGT described in this paper is impacted largely by the time between consecutive calibration periods, thus periods of missing data can result in high uncertainty before and after the data gaps. To address this last point we have added an 8 hour mask, removing periods from the analysis in which 8 or more hours of data are missing. We have updated the figures and the numbers reported in the text and Table 3 to represent this small change.

To address this, language has been added in section 4, line 534, that describes each of these factors. This language reads

"It should be noted that since UpTGT is time dependent, gaps in data will result in large uncertainties estimates. As a result we have added a mask, in which any period of 8 hours or more of data are removed from the UpTGT calculation. Additionally, bias in the assigned values of calibration tanks, as well as changes in the distribution of the mole fraction of calibration tanks on site, can result in result in step wise changes in UpTGT as can be seen if figures 10 and 11."

Please also note the supplement to this comment:
https://www.earth-syst-sci-data-discuss.net/essd-2018-148/essd-2018-148-AC1-supplement.pdf